# Super-resolution microscopy reveals coupling between mammalian centriole subdistal appendages and distal appendages

**Weng Man Chong[1], Won-Jing Wang[2], Chien-Hui Lo[2], Tzu-Yuan Chiu[1], Ting-Jui Chang[1], You-Pi Liu[1], Barbara Tanos[3], Gregory Mazo[4], Meng-Fu Bryan Tsou[5], Wann-Neng Jane[6], T Tony Yang[7,8]\*, Jung-Chi Liao[1]\***

[1]Institute of Atomic and Molecular Sciences, Academia Sinica, Taipei, Taiwan; [2]Institute of Biochemistry and Molecular Biology, National Yang Ming University, Taipei, Taiwan; [3]Division of Cancer Therapeutics, The Institute of Cancer Research, London, United Kingdom; [4]Dermatology Service, Department of Medicine, Memorial Sloan Kettering Cancer Center, New York, United States; [5]Cell Biology Program, Memorial Sloan-Kettering Cancer Center, New York, United States; [6]Institute of Plant and Microbial Biology, Academia Sinica, Taipei, Taiwan; [7]Graduate Institute of Biomedical Electronics and Bioinformatics, National Taiwan University, Taipei, Taiwan; [8]Department of Electrical Engineering, National Taiwan University, Taipei, Taiwan

**Abstract** Subdistal appendages (sDAPs) are centriolar elements that are observed proximal to the distal appendages (DAPs) in vertebrates. Despite the obvious presence of sDAPs, structural and functional understanding of them remains elusive. Here, by combining super-resolved localization analysis and CRISPR-Cas9 genetic perturbation, we find that although DAPs and sDAPs are primarily responsible for distinct functions in ciliogenesis and microtubule anchoring, respectively, the presence of one element actually affects the positioning of the other. Specifically, we find dual layers of both ODF2 and CEP89, where their localizations are differentially regulated by DAP and sDAP integrity. DAP depletion relaxes longitudinal occupancy of sDAP protein ninein to cover the DAP region, implying a role of DAPs in sDAP positioning. Removing sDAPs alter the distal border of centrosomal γ-tubulins, illustrating a new role of sDAPs. Together, our results provide an architectural framework for sDAPs that sheds light on functional understanding, surprisingly revealing coupling between DAPs and sDAPs.

\*For correspondence:
tonyyang@ntu.edu.tw (TTY);
jcliao@iams.sinica.edu.tw (J-CL)

**Competing interests:** The authors declare that no competing interests exist.

## Introduction

The mammalian centrosome is a microtubule organizing center (MTOC) that plays a crucial role in various biological processes, such as mitotic spindle organization and cell polarity regulation (*Oakley et al., 1990*; *Bystrevskaya et al., 1988*; *Bornens, 2012*). A centrosome consists of a pair of rod-shaped centrioles, i.e. a mother centriole and a daughter centriole, and the pericentriolar material (PCM). γ-tubulin is considered as the major microtubule (MT)-nucleating factor of the PCM by forming the γ-tubulin ring complex (γTuRC) as the nucleation template (*Moritz et al., 1995*; *Zheng et al., 1995*; *Pereira and Schiebel, 1997*; *Wiese and Zheng, 1999*). The size of the PCM indicated by the signal of γ-tubulin changes with the cell cycle, from a tightly packed PCM in the

interphase to a large aggregate during mitosis, with the mother and daughter centrioles as the core (*Khodjakov and Rieder, 1999*; *Robbins et al., 1968*; *Sonnen et al., 2012*).

The mother centriole is structurally differentiated from the daughter centriole by the presence of two appendage structures close to its distal end, namely distal appendages (DAPs) and subdistal appendages (sDAPs). DAPs are essential for centriole-membrane docking and primary cilia formation (*Tanos et al., 2013*; *Ye et al., 2014*; *Schmidt et al., 2012*; *Ishikawa et al., 2005*; *Joo et al., 2013*). We and others have identified a number of distal appendage (DAP) proteins, including C2CD3, CEP83, CEP89, SCLT1, FBF1 and CEP164 (*Tanos et al., 2013*; *Ye et al., 2014*; *Graser et al., 2007*; *Wei et al., 2013*; *Sillibourne et al., 2013*; *Joo et al., 2013*). We have also used super-resolution microscopy to map the molecular architecture of mammalian DAPs toward the distal end of the mother centriole (*Yang et al., 2018*). Super-resolved images showed that DAPs are composed of DAP blades of nine-fold symmetry, as observed in electron microscopy (EM), and the novel structure of the DAP matrix between adjacent blades. The Loncarek group further used correlative super-resolution microscopy and EM to show the precise localization of DAP proteins relative to the electron dense structure of DAPs (*Bowler et al., 2019*), improving the architectural mapping of the DAPs. We also found that CEP89 occupies two longitudinal layers, one close to the DAP region and the other close to the putative subdistal appendage (sDAP) region, proximally adjacent to the DAP region shown in a number of EM images of the mother centrioles.

Some EM images showed variations in the number of sDAPs, such as 2 to 12 sDAP stems in human endotheliocytes (*Bystrevskaya et al., 1992*; *Bystrevskaya et al., 1988*), illustrating the dynamic nature of sDAPs. That is, in contrast to an exact number of nine DAPs per centriole, the number of sDAPs may be different in different centrioles (*Uzbekov and Alieva, 2018*). Even when sDAPs do form a complete ring of nine-fold symmetry, their morphology is different from that of DAPs (*Paintrand et al., 1992*; *Uzbekov and Alieva, 2018*). Each sDAP stem is composed of one pair of electron-dense signals on the sides, where one of these signals is associated with the A-tubule of a MT triplet of the mother centriole and the other is associated with the C-tubule of an adjacent MT triplet (*Bystrevskaya et al., 1988*; *Paintrand et al., 1992*). The longitudinal positions of mammalian sDAPs may also vary along the mother centriole. For centrioles of motile cilia, the structure at the longitudinal position proximal to the DAPs is the basal foot, a 'badminton-shaped' structure that is largely different from the sDAPs in the mother centriole (*Anderson, 1972*; *Bornens, 2012*; *Bystrevskaya et al., 1988*). The basal foot is oriented to align with the beating direction of motile cilia, suggesting its role in mechanical coupling or anchoring (*Gibbons, 1961*; *Clare et al., 2014*). We and others have identified a series of sDAP proteins, including ODF2, CEP128, centriolin, CCDC68, CCDC120, ninein and CEP170 (*Mazo et al., 2016*; *Huang et al., 2017*; *Guarguaglini et al., 2005*; *Nakagawa et al., 2001*; *Ou et al., 2002*; *Mogensen et al., 2000*; *Schrøder et al., 2012*). Although the structural presence of sDAPs are routinely observed in EM images, detailed geometrical information to describe the locations of these proteins relative to each other remains largely missing.

Although DAPs and sDAPs are mostly adjacent to each other in mammalian centrioles, their functions are distinct. DAPs are responsible for ciliogenesis and ciliary vesicle docking (*Tanos et al., 2013*; *Schmidt et al., 2012*; *Joo et al., 2013*). sDAPs are mostly considered to serve roles in MT anchoring (*Vorobjev and Chentsov, 1982*; *Bystrevskaya et al., 1988*; *Bornens, 2002*; *Delgehyr et al., 2005*). EM images show that MTs terminate at the tips of sDAPs, but one can also see many other MTs around the centrioles (*Vorobjev and Chentsov, 1982*; *Mogensen et al., 2000*). Direct functional evidence remains to be shown. Centriole MT aster reformation after MT depolymerization is delayed in ninein or CEP170 knockout or knockdown cells (*Delgehyr et al., 2005*; *Guarguaglini et al., 2005*). However, both ninein and CEP170 have two localization populations, one at the centriole proximal end, and the other in the region close to sDAPs (*Sonnen et al., 2012*). Thus, it cannot be confirmed whether the phenotypes are due to abolished sDAPs or to a damaged centriole proximal end. It is also unclear whether the MT anchoring of the sDAPs requires γ-tubulin or γTuRC as a nucleating factor. Other functional roles of sDAPs have also been reported. We have previously shown that, despite being dispensable for ciliogenesis, depletion of sDAPs together with the loss of the centriole proximal end protein CNAP1 results in the detachment of cilia and Golgi, affecting the positioning of primary cilia (*Mazo et al., 2016*). Another study on sDAP protein CEP128 found that CEP128 is implicated in RAB11 vesicle trafficking at the primary cilia, in which

CEP128 depletion results in a defect in TGF-β/bone morphogenetic protein (BMP) signaling and abnormal organ development in zebrafish (*Mönnich et al., 2018*).

Studies of the DAP–sDAP relationship are also very limited. As mentioned above, our previous super-resolution imaging result showed dual localizations of CEP89, potentially one localization in the DAP region and the other close to the sDAP region, but no functional studies have been performed to check whether CEP89 is associated with sDAPs or not. In RPE-1 cells, a genetic knockout of the most upstream sDAP protein ODF2 was shown to abolish the recruitment of other sDAP components (*Mazo et al., 2016*), but neither the DAP assembly (*Tanos et al., 2013*) nor the ability of the mutant centrioles to grow cilia (*Mazo et al., 2016*) was grossly impacted. Similarly, reduction of an upstream DAP protein, CEP83 in CEP83 mutated fibroblasts, did not affect the localization of ODF2 (*Failler et al., 2014*), suggesting that DAPs and sDAPs are two independent elements. However, separate studies showed that a null ODF2 knockout in the mouse F9 cell line abolished both the distal and subdistal appendages (*Ishikawa et al., 2005*), and that partial truncation of ODF2 could abolish sDAPs without affecting DAPs (*Tateishi et al., 2013*), revealing a more complicated relationship between the two structures. Together, despite their proximity to each other, whether distal and subdistal appendages are structurally or functionally related to each other remains elusive.

In this study, we systematically characterized the super-resolved molecular architecture of sDAPs using direct stochastic optical reconstruction microscopy (dSTORM). We mapped the spatial organization of multiple sDAP-related proteins and generated a three-dimensional model. Intriguingly, we found structural coupling between DAP and sDAP proteins. We have also directly observed the role of sDAPs on MT anchoring and further illustrated the selective involvement of γ-tubulins on MT templating. Together, our studies systematically reveal the super-resolved architecture of sDAPs, suggest potential DAP–sDAP structural connections, and implicate a regulatory mechanism of MT anchoring around the centrosome.

## Results

### Super-resolution microscopy reveals that the completeness of ring occupancies by sDAP proteins depends on the cell cycle

To characterize sDAP architecture systematically, we imaged multiple sDAP proteins reported previously (*Mazo et al., 2016*; *Huang et al., 2017*; *Tateishi et al., 2013*; *Mönnich et al., 2018*) by dSTORM super-resolution microscopy. ODF2, CEP128, centriolin, CCDC68, ninein and CEP170 of sDAPs were imaged. A centriole proximal end protein CNAP1, known to be associated with some sDAP proteins (e.g. ninein, CEP170), was also imaged (*Mazo et al., 2016*). It is known that the organization of centriole proteins may change with cell-cycle phases (*Guarguaglini et al., 2005*). Here, we only focus on the geometric arrangement of proteins in the quiescent state, where primary cilia are grown or ready to grow. RPE-1 cells were serum starved for at least 24 hr to synchronize cells to the $G_0$ phase before proceeding to imaging.

We first examined whether sDAP proteins form rings with a nine-fold symmetry like those formed by DAP proteins. We conducted two-color staining of the DAP protein SCLT1 and an sDAP protein of interest using dSTORM with Alexa Fluor 647 and Cy3B organic dyes. We used SCLT1 as a marker in widefield imaging to pick centrioles approximately viewed from the axial view by means of the well-defined ring structure of SCLT1 that is crudely observable in conventional widefield microscopy (*Figure 1a,d*). We then imaged the channel for an sDAP protein of interest using super-resolution microscopy and examined the radial distributions of the sDAP protein. Interestingly, we found different angular integrities of sDAP proteins among the images of the same protein as well as among those of different sDAP proteins (*Figure 1a and d*). As shown in *Figure 1a*, the majority of CEP170 rings are not intact. Statistically, CEP170 exhibits a wider range of radial distributions under serum-fed (cell proliferating) conditions than under serum-starved (cell resting) conditions (*Figure 1b and c*). In contrast to CEP170, the relatively upstream sDAP protein CEP128 is better organized during the cell-resting phase and during the proliferating phase (*Figure 1d*). Circular expansion of signals along the polar coordinate of the ring illustrates that ~50% of ring occupancy is missing under serum-fed conditions, whereas ~25% is missing under serum-starved conditions (*Figure 1e and f*). It is interesting to note that 24-hr serum starvation, which promotes ciliogenesis in around 60% of RPE-1 cells (*Lo et al., 2019*), does have an effect on the completeness of ring occupancy of

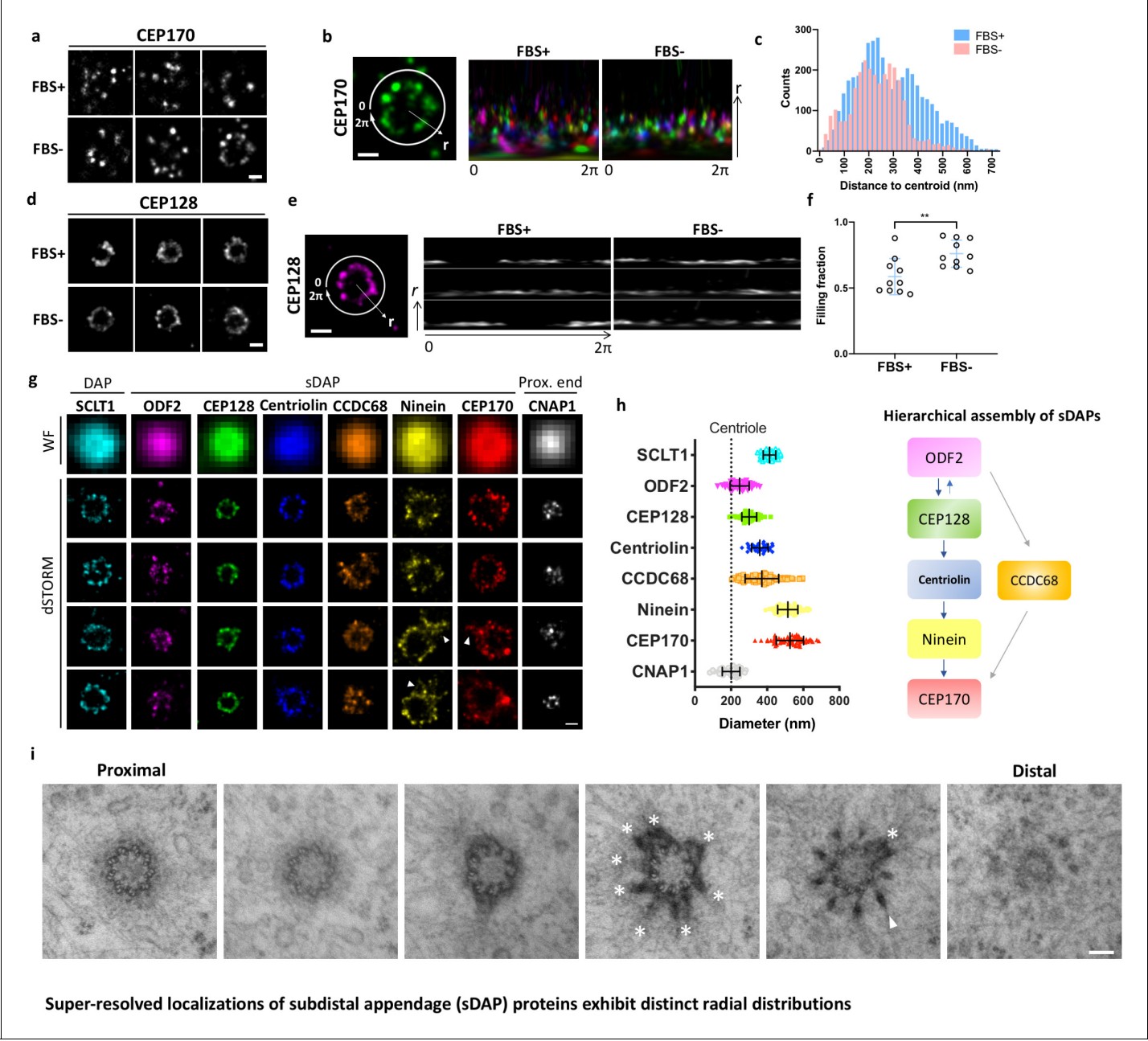

**Super-resolved localizations of subdistal appendage (sDAP) proteins exhibit distinct radial distributions**

**Figure 1.** Super-resolved localizations of subdistal appendage (sDAP) proteins exhibit distinct radial distributions. (a) Axial-view direct stochastic optical reconstruction microscopy (dSTORM) images of CEP170 under serum-supplied (FBS+) and serum-starved (FBS–) conditions. (b) Image analysis revealing that, under the proliferating condition (FBS+), CEP170 exhibits a relatively random radial distribution compared to that under the resting (FBS–) G$_0$ phase. Each color represents a data value for different centrioles. (c) Statistical analysis of the radial distribution of CEP170 under FBS+ and FBS– conditions. n = 10 centrioles for both conditions, p<0.05. (d) Axial-view dSTORM images of CEP128 under serum-supplied (FBS+) and serum-starved (FBS–) conditions. (e) Image analysis revealing that CEP128 rings are less organized under the proliferating condition (FBS+) than under the resting phase (FBS–). (f) Statistical analysis showing the completeness of the ring-shaped occupancy of CEP128 under FBS+ and FBS– conditions. **p<0.01. (g) Representative dSTORM super-resolution images of axial-view centrioles showing the radial distribution of the distal appendage (DAP) protein SCLT1, various sDAP proteins, and the centriole proximal-end (Prox. end) protein CNAP1, which were not resolvable under wide-field (WF) imaging. 'Overhang structures' (arrowheads) were sometimes observed in the ninein and CEP170 rings. (h) (Left) Mean diameter analysis revealing size differences among the proteins in panel (g). *Supplementary file 1* lists the dimensional details. The diameters of ODF2 and CNAP1 were similar to that of the centriole wall measured from the electron microscopy (EM) images (dotted line). (Right) A schematic figure summarizing previous studies on the hierarchical assembly of sDAPs. (i) Serial transmission EM (TEM) sections of an RPE-1 mother centriole. The centriole, reconstituted by TEM analysis with serial sectioning, reveals an approximately nine-fold distribution of sDAP in RPE-1 cells. Asterisks and the arrowhead indicate sDAPs and DAP, respectively. Bars: panels (a, b, d, e, g) = 200 nm; panel (i) = 100 nm.

*Figure 1 continued on next page*

*Figure 1 continued*

The online version of this article includes the following figure supplement(s) for figure 1:

**Figure supplement 1.** Localization of CEP170 using 3D expansion microscopy.

ciliogenesis-independent sDAP proteins. In addition, sDAPs may not always possess a nine-punctum pattern like that of DAPs in RPE-1 cells. Our results not only show inconsistent occupancies of sDAPs, but also demonstrate a variation in sDAP ring completeness, probably in different cell phases, in the presence or absence of serum. Furthermore, our super-resolution fluorescence study enables us to discriminate occupancy variations of an upstream sDAP protein (CEP128) and of a peripheral sDAP protein (CEP170).

## ODF2 is close to the centriole microtubule wall whereas ninein and CEP170 are close to the sDAP tips

To understand the architecture of sDAP proteins with nearly full occupancies at the sDAPs, we pre-selected centrioles based on whether we saw a ring or not for an sDAP protein of interest under widefield microscopy, and further analyzed these centrioles with super-resolution microscopy. As shown in *Figure 1g and h*, the sDAP proteins are in ring shapes of different sizes, ranging approximately from 250 nm to 600 nm in diameter. Among them, ODF2 forms the smallest ring with a diameter similar to that of the centriole MT wall (~200 nm), suggesting its close proximity to the centriole. By contrast, ninein and CEP170 occupy a much wider space, with a diameter of around 600 nm, larger than the diameter of outer DAP proteins such as FBF1 and SCLT1. Multiple studies have characterized the spatial organization of sDAP proteins via immuno-EM (*Guarguaglini et al., 2005*; *Huang et al., 2017*) or structured illumination (SIM) microscopy (*Sonnen et al., 2012*; *Huang et al., 2017*; *Kashihara et al., 2019*). In line with previous SIM studies, our dSTORM images also show a range of 250–600 nm in diameter for various sDAP proteins. The difference is that SIM is limited by the ~100 nm lateral imaging resolution, whereas dSTORM can achieve a resolution of 20 nm. Taking advantage of the resolving power of dSTORM imaging, our work is able to map the molecular architecture in more detail.

When correlating the occupancies of sDAP proteins from ODF2 to CEP170 to the reported assembly hierarchy of sDAPs as illustrated in *Figure 1h* (*Mazo et al., 2016*; *Huang et al., 2017*), we found that the smaller the size of the protein distribution, the more upstream it is in the hierarchy. Occasionally, an overhang-like structure (*Figure 1g*, marked by arrowhead) was observed for ninein and CEP170. To better understand whether this overhang-like structure is a basal foot-like protrusion or a part of a neighboring daughter centriole, we performed 3D two-color super-resolution imaging of CEP170 and a DAP protein CEP164 using expansion microscopy (*Figure 1—figure supplement 1A*). When looking at these images together with conventional epifluorescent microscopy imaging (*Figure 1—figure supplement 1B*), one can see that the extra signal most probably comes from the proximal end localization of CEP170 on the daughter centriole, and not from an overhang structure of the sDAPs. To further investigate the arrangement of the sDAP stems, we performed TEM sectioning of RPE-1 centrioles in the axial view, as shown in *Figure 1i*. The TEM images reveal that sDAP stems can be mostly observed in the same section with a nine-fold arrangement. It is also interesting to note that the sDAP stems are mostly wider than those of DAPs, similar to the observations recorded in previous work by the Bornens group (*Paintrand et al., 1992*). From the EM image, it seems that the stems of sDAPs are more flexible in shape and distribution than those of DAPs, potentially explaining the finding that nine-fold distribution is less obvious for the localization of sDAP proteins.

## Lateral super-resolution images reveal sDAP as a triangular structure

We defined the longitudinal position of sDAP proteins by means of dual color dSTORM super-resolution imaging. Each sDAP protein was immuno-stained together with the DAP protein SCLT1, which served as a position reference. As shown in *Figure 2a and b*, CEP128 forms a compact layer at about 160 nm proximal to the SCLT1 layer of the DAPs. ODF2 covers a broader longitudinal region with two populations separating in two layers, which respectively localize at ~100 nm and ~200 nm proximal to the DAP protein SCLT1. Centriolin also covers a longitudinal range that seemingly

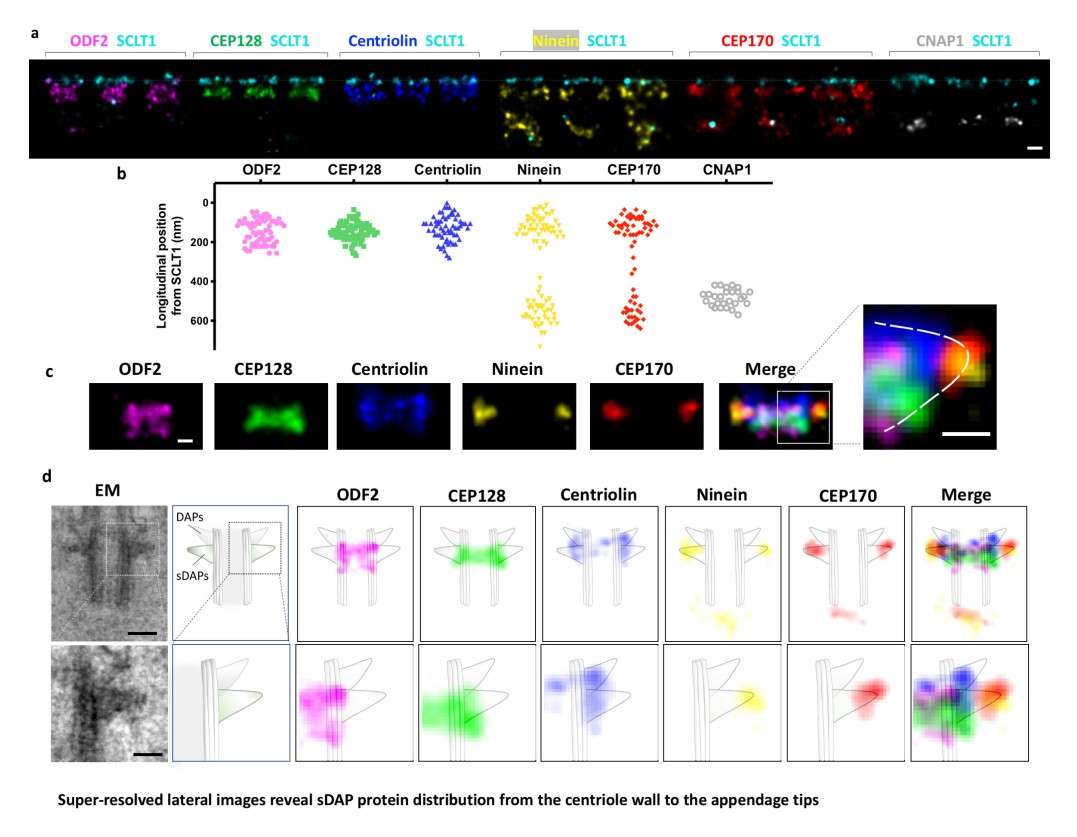

**Figure 2.** Super-resolved lateral images reveal sDAP protein distribution from the centriole wall to the appendage tips. (a) Representative lateral-view two-color dSTORM images of sDAP–SCLT1 pairs revealing the longitudinal positions of various sDAP proteins relative to SCLT1. (b) A scatter plot describing the longitudinal positions of sDAP proteins relative to SCLT1. (c) dSTORM images of each sDAP protein were aligned using SCLT1 as a reference and combined into a composite image (n > 7 centrioles each). The composite images were then aligned and merged according to their average longitudinal position. (Inset) A magnified image showing the triangular-like arrangement of the sDAP structure (dashed line). (d) (Top panel) Composite dSTORM images of each sDAP protein in panel (c) overlaid with a mother centriole cartoon model, which is depicted from the TEM image of RPE-1 cells (left) to illustrate the potential localization of sDAPs on the sDAP stems. (Bottom panel) Magnified view of the inset in panel (d). Bars: panels (a, d) (top) = 200 nm; panel (c) (inset), (d) (bottom) = 100 nm.

possesses a single layer, although we cannot rule out the possibility of two very close layers. Ninein and CEP170 localize both to the sDAP region and to the centriole proximal end marked by CNAP1. The two layers are separated by ~350 nm. *Figure 2c* shows the alignment of sDAP proteins according to their longitudinal positions. CEP128 is sandwiched between the two ODF2 layers, with centriolin localized above them. Ninein and CEP170 localize radially outward from centriolin. The composite image of these sDAP proteins shows a left-right symmetric structure and each unit is triangular in shape. A schematic model for the positioning of sDAP proteins is illustrated in *Figure 2d*. The cartoon of the centriole is constructed on the basis of an EM side view image of RPE-1 cells in which the scale is adjusted to be comparable to that of the dSTORM images. Assuming that CEP170 is localized at the tip of sDAP, as observed in the previous immune-EM study (*Guarguaglini et al., 2005*), ODF2 is located at the centriole wall and its two layers occupy the two ends of the sDAP root, that is the portion of an sDAP stem connecting to the centriole wall. Other sDAP proteins fill up the triangular structure. CEP128 is right at the border between the centriole MT and the sDAP stem. Centriolin is located at the middle portion of an sDAP stem, whereas ninein is close to the tips of sDAPs.

## Axial and lateral images compose a 3D molecular map of sDAPs

Combining sDAP measurements from our axial and lateral dSTORM images (*Figure 3a*), we generated a 3D molecular model illustrating the localization of various sDAP proteins on a centriole model

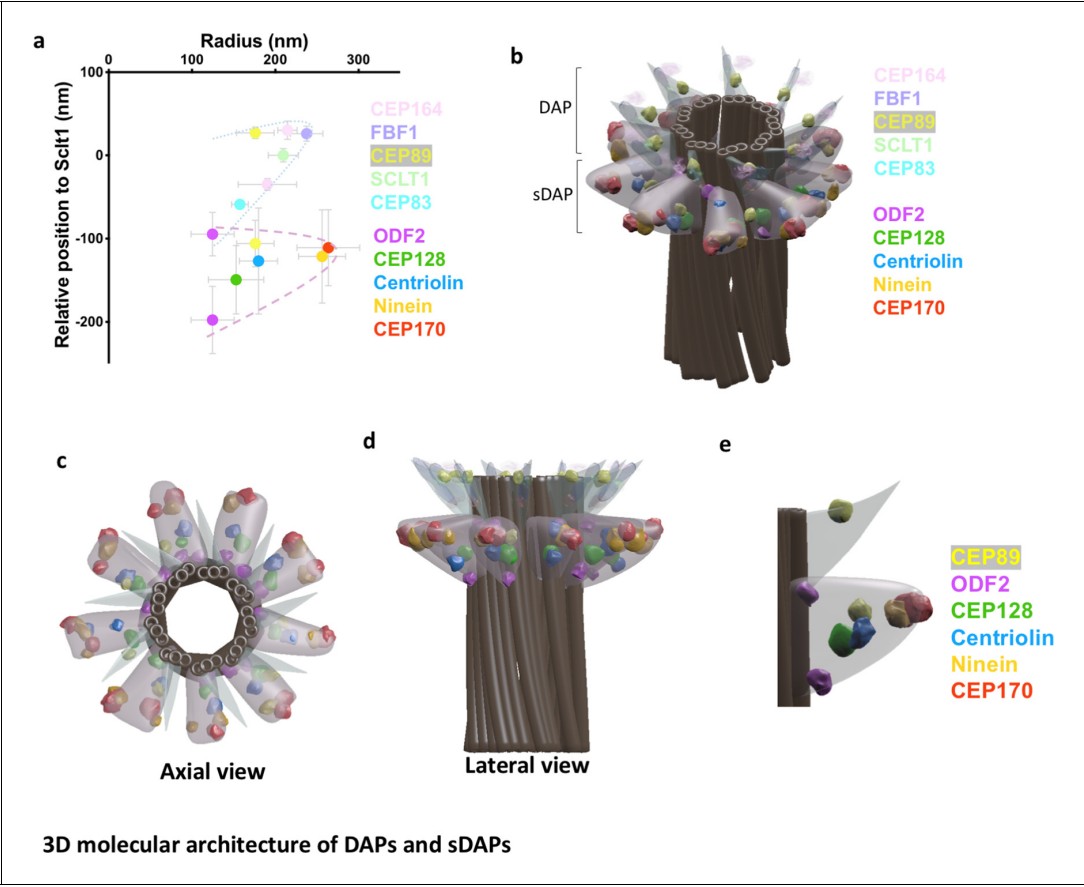

**Figure 3.** 3D molecular architecture of DAPs and sDAPs. (a) Relative localization of DAP and sDAP proteins in radial and lateral directions revealing the slanted arrangement of a DAP (dotted line) and the triangular structure of an sDAP (dashed line). (b) A 3D model of a mother centriole, illustrating the localization of various sDAP and DAP proteins for one of the possible arrangements when all nine sDAPs are present. (c) An axial view of the 3D model in panel (b) viewed from the distal end of the centriole, illustrating the radial positions of CEP89 and various sDAP proteins. (d) Lateral view of the model in panel (c). (e) Close view of the sDAP and the DAP in panel (d). ODF2 localizes at both ends of the sDAP and close to the centriole wall; CEP89 localizes on the DAP as well as in the sDAP region.

(*Figure 3b*, *Video 1*). The model of the centriole and the DAPs is based on previous TEM serial sections of the monkey oviduct basal body (*Anderson, 1972*); the sDAP model is constructed on the basis of the TEM study of centrioles from a human lymphoblastoma cell line (*Paintrand et al., 1992*) and our sDAP TEM results (*Figure 1i*). Note that because the structures of sDAPs are dynamic (*Uzbekov and Alieva, 2018*), this model only represents one possible organization of sDAPs in a subset of centrioles, such that other settings may also exist. ODF2, CEP128 and centriolin lie close to the centriole MT wall. CEP170 and ninein are mostly at the sDAP tips. To gain further structural insight, we also included the DAP structure from our previous work in our model (*Figure 3b–e*). Note that both the sDAP protein ODF2 and the DAP protein CEP89 are two

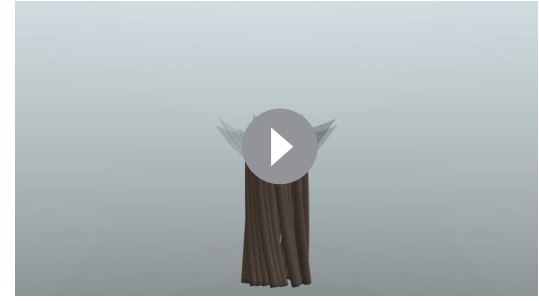

**Video 1.** 3D view of a mother centriole model containing (i) DAP and sDAP proteins, (ii) a 3D molecular model of a mother centriole with reference to the centriole, sDAP, and DAP from previous centriole, and (iii) TEM results (*Anderson, 1972*; *Paintrand et al., 1992*) and our sDAP TEM results (*Figure 1i*), together with sDAP results from axial and lateral dSTORM images (*Figure 3a*).
https://elifesciences.org/articles/53580#video1

layered structures (*Figure 3d,e*). The proximal layer of ODF2 is at the lowest position among the sDAP proteins studied; while the distal layer of ODF2 lies in close proximity to the core DAP protein CEP83 (i.e. ~60 nm proximal to SCLT1 as we previously reported; *Yang et al., 2018*). The distal layer CEP89 lies at the distal end of DAPs; while its proximal layer is proximal to CEP83 and lies in the sDAP region. These observations suggest a potential structural relationship between sDAPs and DAPs.

## ODF2 and CEP89 localizations are differentially regulated by DAP and sDAP integrity

To understand the relationship of ODF2 and DAPs/sDAPs, we conducted DAP depletion (i.e. CEP83 CRISPR/Cas9 knockout) and sDAP depletion (i.e. CEP128 CRISPR/Cas9 knockout) to check their effects on ODF2 localization with dSTORM. Successful knockout of CEP83 or CEP128 in RPE-1 cells were confirmed by immunoblotting (*Figure 4a*). Lateral super-resolution images of ODF2 reveal that ODF2 has two localizations distal and proximal to CEP128 in wild-type cells (*Figures 2c* and *4b*). Surprisingly, in DAP or sDAP knockout cells, single-layered ODF2 was observed (*Figure 4b and e*). In particular, the proximal layer of ODF2 was absent upon CEP128 depletion, where SCLT1 was used as a reference of the longitudinal position. To examine whether this change in ODF2 is caused by a potential change in ODF2 configuration upon CEP128 depletion as reported previously (*Kashihara et al., 2019*), in addition to the original antibody targeting the N terminus of ODF2 (aa #39–200) (denoted as ODF2-N in *Figure 4b*), we used another antibody targeting the C terminus of ODF2 (aa #800 to the C terminus, denoted as ODF2-C) for further study (*Figure 4c*). dSTORM imaging of ODF2-C revealed that in wild-type cells, ODF2-C distribution is wider than ODF2-N distribution in the radial direction (*Figure 4b, c and d*). In addition, the distal edge of ODF2-C is closer to SCLT1 than that of ODF2-N, reaching further toward the centriole distal end. Interestingly, when CEP128 is depleted, distributions of both ODF2-N and ODF2-C become thinner. The gap between ODF2-N/C and SCLT1 of CEP128$^{-/-}$ cells is larger than that of the wild-type cells, illustrating both narrowing and shifting of ODF2 occupancy upon sDAP CEP128 depletion. These observations imply that CEP128, as the binding partner of ODF2 (*Kashihara et al., 2019*), regulates the organization of ODF2. Whether this ODF2 thin layer in CEP128$^{-/-}$ cells is originated from the ODF2 layer at DAP region or the one at sDAP region remains to be examined. One speculation is that the sDAP depletion results in the removal of ODF2 in the sDAP layer. Similarly, in CEP83-depleted centrioles, one layer of ODF2 was absent (*Figure 4e*). This observation implies that ODF2 is associated with both DAPs and sDAPs as a downstream protein of CEP83 and CEP128, respectively. Previous studies on the role of ODF2 with regard to DAPs and sDAPs have been controversial (*Tanos et al., 2013*; *Ishikawa et al., 2005*; *Tateishi et al., 2013*). Our results structurally confirm that ODF2 is involved in hierarchical relationships with both DAP and sDAP proteins (*Tateishi et al., 2013*; *Mazo et al., 2016*). Note that our finding here of one layer of ODF2 as a downstream protein of CEP128 explains our previous genetic studies showing reduced ODF2 levels due to CEP128 depletion (*Mazo et al., 2016*).

We also examined the sDAP association of CEP89, a putative DAP protein that also exhibits a two-layered localization, by checking the effect of CEP89 localization upon sDAP depletion (CEP128 CRISPR/Cas9 knockout). Intriguingly, the proximal layer of CEP89 was absent when sDAP was depleted (*Figure 4f and g*). While DAP was depleted, CEP89 signal was reduced as compared to that in the wild type cells (*Figure 4—figure supplement 1*). That is, similar to ODF2, CEP89 has dual roles associated with both DAPs and sDAPs. These findings suggest an interesting structural–functional interplay between DAP and sDAP structures.

## Ninein covers a broadened longitudinal region toward the centriole distal end upon DAP removal

To further examine the interaction of sDAPs and DAPs, we checked whether ninein at sDAPs is affected by DAP depletion. Surprisingly, super-resolution images show that in DAP-depleted *CEP83$^{-/-}$* cells, ninein at the sDAPs covers a longitudinal region larger than that in wild-type cells (*Figure 4g and h*). Using CP110 as a reference to indicate the distal boundary of the centriole, we find that the broadened longitudinal region of ninein spans from the sDAP region to the region originally occupied by DAPs. In addition, when CEP83 is stably expressed in *CEP83$^{-/-}$* cells (*CEP83$^{-/-}$*

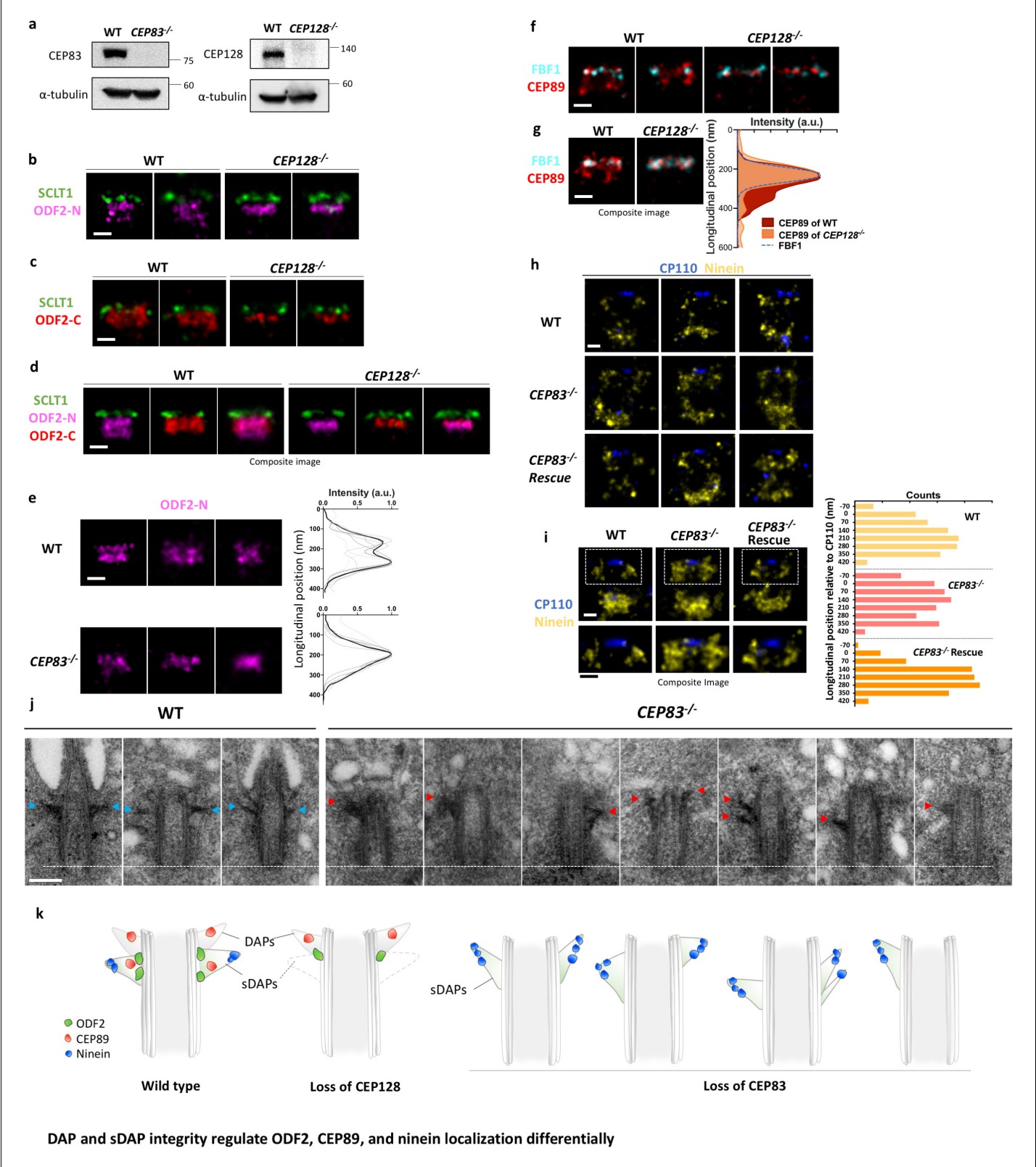

**Figure 4.** DAP and sDAP integrity regulates ODF2, CEP89, and ninein localization differentially. (a) Immunoblotting confirming knockout of CEP83 or CEP128 in RPE-1 cells. WT, wild-type RPE-1 cells; *CEP128⁻/⁻*, CEP128 knockout RPE-1 cells. (b, c) Representative two-color dSTORM images of (b) the N-terminus of ODF2 and (c) the C-terminus of ODF2 with SCLT1 in WT cells and *CEP128⁻/⁻* cells. (d) Two-color dSTORM images in panels (b) and (c) aligned and combined according to their longitudinal positions relative to SCLT1 (n = 5 centrioles per group). (e) (Left) Representative dSTORM images
*Figure 4 continued on next page*

Figure 4 continued

of ODF2 in WT and CEP83 knockout RPE-1 cells (*CEP83⁻/⁻*). (Right) Intensity profile of WT and *CEP83⁻/⁻* cells (WT, n = 7 centrioles; *CEP83⁻/⁻*, n = 6 centrioles) showing that ODF2 becomes a single-layer structure when CEP83 is depleted. (f) Representative two-color dSTORM images of CEP89 and FBF1 in WT and *CEP128⁻/⁻*cells. (g) (Left) Two-color dSTORM images in panel (e) aligned and combined according to their longitudinal positions relative to FBF1 (WT, n = 5 centrioles; *CEP128⁻/⁻*, n = 6 centrioles). (Right) Intensity profile of the images in the left panel showing that the lower layer of CEP89 is absent in the *CEP128⁻/⁻* cells. (h) Representative two-color dSTORM images of ninein and CP110 in WT, *CEP83⁻/⁻*, and *CEP83⁻/⁻* cells stably expressing wild-type CEP83 protein (*CEP83⁻/⁻Rescue*). (i) (Left, top) Two-color dSTORM images in panel (g) aligned and combined according to their longitudinal positions relative to CP110 (WT, n = 5 centrioles; *CEP83⁻/⁻*, n = 6 centrioles; *CEP83⁻/⁻Rescue*, n = 5 centrioles). (Left, bottom) Magnified images of the insets in the top row. (Right) Histograms for the images on the left revealing that ninein is distributed towards the centriole distal end in *CEP83⁻/⁻* cells as compared to that in WT and *CEP83⁻/⁻Rescue* cells. (j) TEM of the mother centriole of WT and *CEP83⁻/⁻*RPE-1 cells. sDAPs in the WT and *CEP83⁻/⁻*cells are marked by blue and red arrowheads, respectively. (k) Cartoon model illustrating the changes in sDAP protein localization upon CEP128 depletion and the variations of sDAP structure upon CEP83 depletion. Bars = 200 nm.

The online version of this article includes the following figure supplement(s) for figure 4:

**Figure supplement 1.** CEP89 intensity is significantly reduced upon CEP83 knockout.

Rescue), the sDAP distribution of ninein is restored (*Figure 4h and i*). To examine the differences in sDAP structures between wild type and CEP83⁻/⁻ RPE-1 cells, we performed TEM with the results shown in *Figure 4j*. In wild-type cells, sDAPs mostly maintained their longitudinal position close to and proximal to the DAPs, but their shapes varied in different centrioles or even in the same centriole (*Figure 4j*, left panel). Surprisingly, we found that the positions and shapes of the stems of sDAPs are largely varying in CEP83-depleted cells were highly variable (*Figure 4j*, right panel). That is, the sDAP structure is much less stable when DAPs are missing. Specifically, a few examples show that sDAP tips may point toward the distal end with a thicker stem, consistent with what we observed in ninein super-resolution images of CEP83⁻/⁻ cells (*Figure 4h and i*). In other cases, sDAPs can localize at the middle region of the centriole, occupy the same side of the centriole, miss one side of occupancy, or localize at different longitudinal positions in the same centriole. These large variations imply that the loss of DAPs via CEP83 depletion affects the structural stability of sDAPs, providing further evidence that DAPs and sDAPs are structurally related to each other (*Figure 4k*). Note that laterally oriented centrioles were preselected by the pattern of centrin and ninein under widefield imaging prior to dSTORM imaging; therefore, unlike the asymmetry of sDAP stems observed in the EM images of CEP83⁻/⁻ cells (*Figure 4j*, right panel), asymmetry of ninein is less obvious in the ninein super-resolution images of CEP83⁻/⁻ cells (*Figure 4h*).

## sDAP depletion eliminates the anchoring of a subset of microtubules around the mother centriole

Previous studies have shown that the absence of ninein reduced microtubule (MT) regrowth after nocodazole-induced MT depolymerization (*Delgehyr et al., 2005*). However, it is unclear whether this outcome results from the ninein population of the sDAP region or that of the centriole proximal end. To validate whether sDAPs do play a MT-anchoring role, we examined α-tubulin localization in wild-type and *CEP128⁻/⁻* RPE-1 cells during the serum-starved $G_0$ phase. In widefield images, no observable difference is found in terms of centrioles serving as the center of MT networks (*Figure 5a and b*). Many MT fibers still originate from the centrioles in the sDAP-depleted cells, and it is not obvious that sDAPs serve as a MT-anchoring site. An axial view of MTs around the mother centriole obtained using super-resolution microscopy can sometimes show reduced MT numbers in *CEP128⁻/⁻*cells (*Figure 5c and d*), but it remains challenging to confirm whether the effect is due to sDAP depletion. Excitingly, by overlaying multiple lateral super-resolved α-tubulin images of wild-type and *CEP128⁻/⁻*cells, we are able to find a void region of MTs in the sDAP region of *CEP128⁻/⁻* cells (*Figure 5e, f and g*). While previous EM studies have revealed the proximity of MTs and sDAPs (*Mogensen et al., 2000*), this is the first evidence from high-resolution imaging to show that loss of sDAPs results in loss of MT attachments. It illustrates a direct role for sDAP in MT anchoring. In addition to sDAP's role in anchoring MTs, our lateral-view images also show another subset of MTs that terminate at the DAP region (*Figure 5e and f*). These DAP-associated MTs do not seem to be affected by sDAP depletion.

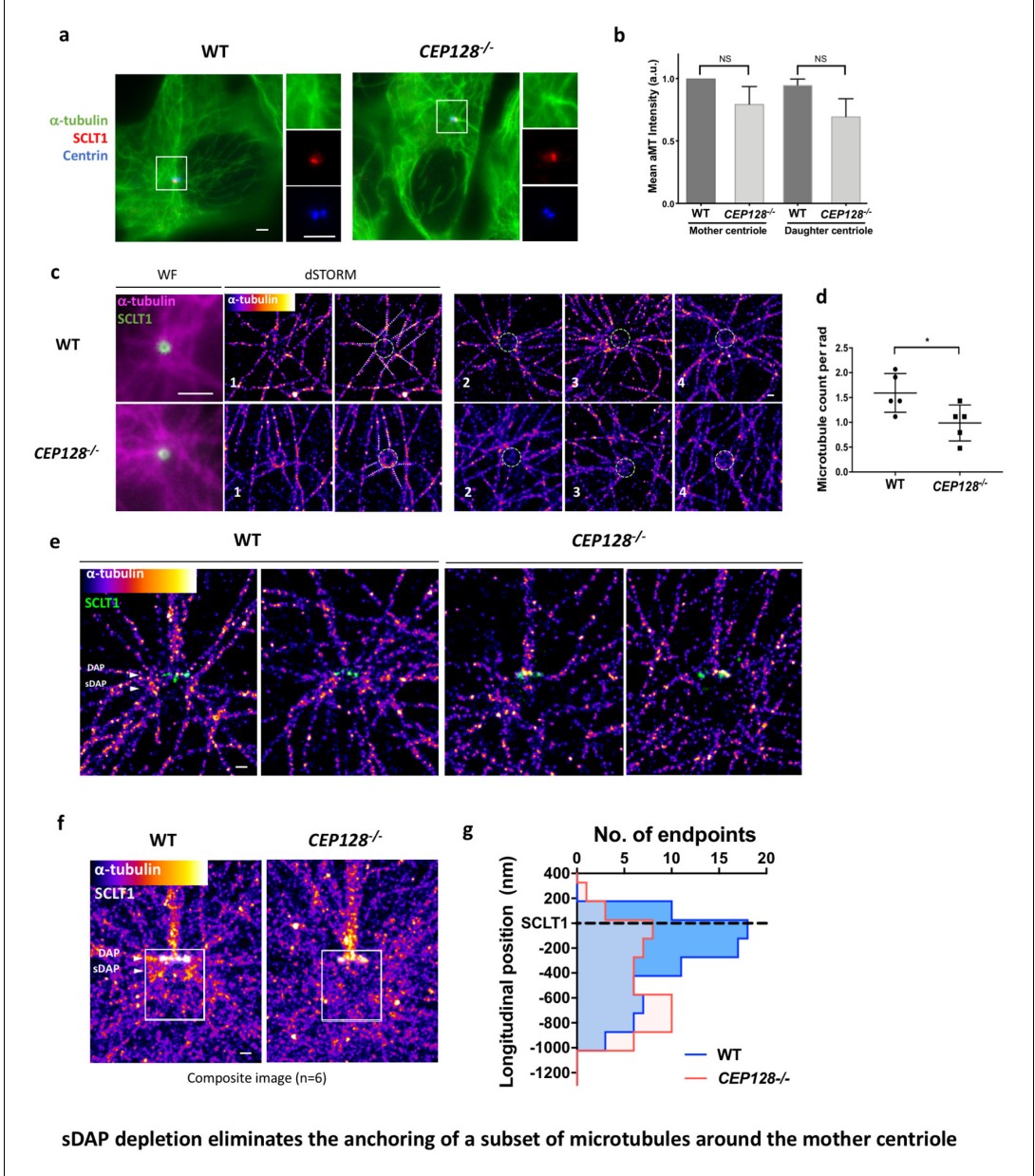

**sDAP depletion eliminates the anchoring of a subset of microtubules around the mother centriole**

**Figure 5.** sDAP depletion eliminates the anchoring of a subset of microtubules around the mother centriole. (**a**) Representative immunofluorescence images of α-tubulins in WT and $CEP128^{-/-}$ RPE-1 cells. Centrin was the centriole marker whereas SCLT1 was the mother-centriole-specific marker. (Inset) Magnified images showing the organization of α-tubulin fibers around the centriole. (**b**) Analysis of mean α-tubulin intensity around mother and daughter centrioles in WT and $CEP128^{-/-}$ cells. NS, not significant. (**c**) (Left) dSTORM images revealing fewer α-tubulin fibers arranged around the $CEP128^{-/-}$ centriole (white dotted line) as compared to the WT centriole. (Right) Representative dSTORM images of α-tubulin fibers in WT and $CEP128^{-/-}$centrioles. SCLT1 is co-stained as a marker for the axial centriole view (green dashed line). (**d**) Statistical analysis counting the number of α-tubulin fibers per radian (rad) around the WT and $CEP128^{-/-}$centrioles (n = 5 centrioles each), *p<0.05. *Figure 5—figure supplement 1* details the analysis approach. (**e**) Representative lateral dSTORM images revealing the organization of centrosomal α-tubulin fibers around the WT and $CEP128^{-/-}$centrioles in the longitudinal direction. (**f**) dSTORM images in panel (**e**) aligned and combined using SCLT1 as a position reference. (**g**) Statistical analysis counting the number of α-tubulin fibers at various longitudinal positions around the WT and $CEP128^{-/-}$centrioles (WT, n = 78 data points; $CEP128^{-/-}$, n = 57 data points, p<0.01). *Figure 5—figure supplement 2* shows the analysis strategy. Bars: (a, c) (WF images) = 2 μm; (c, e, f) = 200 nm.

The online version of this article includes the following figure supplement(s) for figure 5:

**Figure supplement 1.** Measuring the number of microtubules around sDAPs in axial view.
**Figure supplement 2.** Measuring the longitudinal position of the microtubule fiber endpoints.

## The majority of γ-tubulins around the mother centriole are not associated with microtubule anchoring at sDAPs in the G$_0$ phase

As γ-tubulin of γTuRC is considered as the nucleation template of MTs, we examined whether γ-tubulin serves this role at the tips of sDAPs for the MTs anchored there. As expected, γ-tubulins are highly enriched close to the centriole (*Figure 6a*). Intriguingly, super-resolution imaging shows that most γ-tubulins cover the longitudinal region of the mother centriole from the proximal end to the height of the sDAPs (*Figure 6a*), leaving the region between the DAPs and sDAPs less populated. The diameter of the γ-tubulin occupied region is about 400 nm, larger than the diameter of the centriole (*Figure 6b*). That is, γ-tubulins in the G$_0$ phase are mostly confined to a well-defined cylindrical position outside the mother centriole. This confined cylindrical distribution of γ-tubulins provides a better-defined localization of the previously known tightly packed PCM in the interphase, which is very different from the broad distribution of PCM-bound γ-tubulins during mitosis. When compared to the distribution of α-tubulin of wild-type centrioles shown in *Figure 5f*, the distal α-tubulin enrichment close to the sDAP region is different from the relatively uniform longitudinal distribution of γ-tubulin. Therefore, it is likely that there are at least two populations of γ-tubulins, a major population that does not anchor α- and β-tubulins and localizes along the mother centriole, and the other associated with MT anchoring at the sDAP region. To further examine the localization

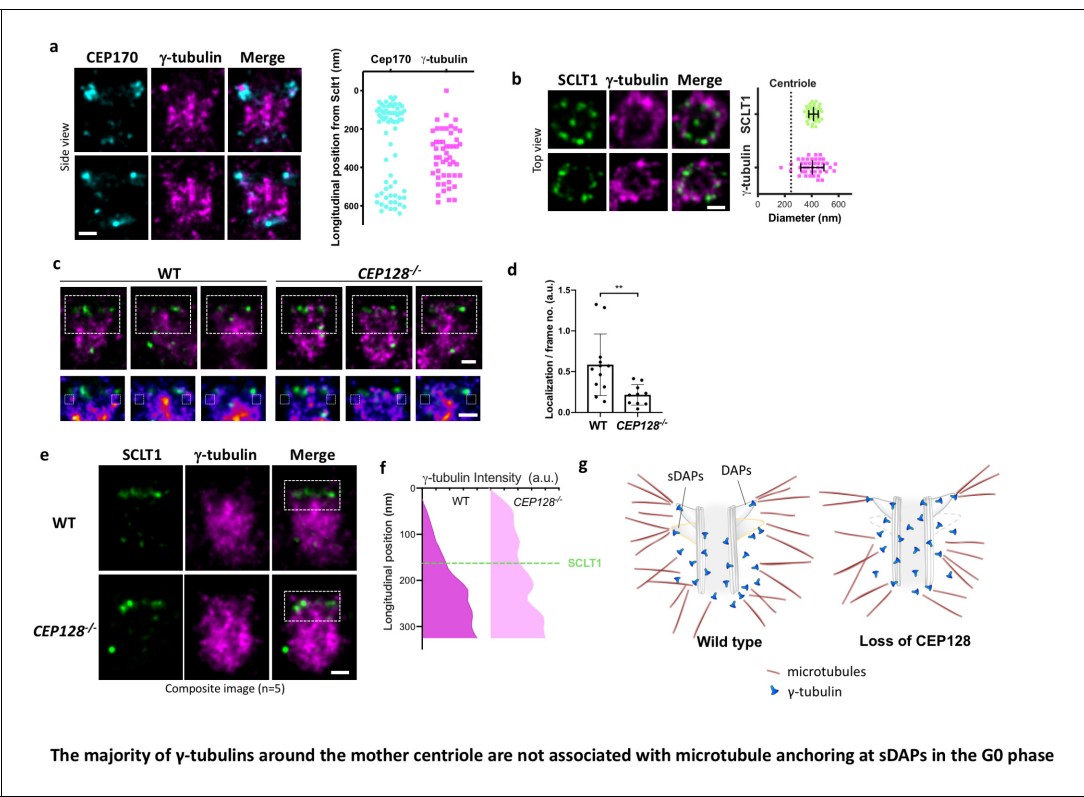

**The majority of γ-tubulins around the mother centriole are not associated with microtubule anchoring at sDAPs in the G0 phase**

**Figure 6.** The majority of γ-tubulins around the mother centriole are not associated with microtubule anchoring at sDAPs in the G$_0$ phase. (**a**) (Left) Representative dSTORM images revealing the longitudinal positions of γ-tubulins with respect to CEP170. (Right) Scatter plot comparing the longitudinal positions of γ-tubulin with that of CEP170 (n > 7 centrioles); the longitudinal position of SCLT1 is set as zero. (**b**) (Left) Representative dSTORM images revealing the radial distribution of γ-tubulins. SCLT1 is the marker for an axial centriole view. (Right) Mean diameter analysis revealing that the radial distribution of γ-tubulins is similar to that of SCLT1. The dotted line indicates the diameter of the centriole wall measured in EM images. (**c**) (Top) Representative lateral two-color dSTORM images revealing the organization of centrosomal γ-tubulin in the WT and *CEP128*$^{-/-}$ centrioles. (Bottom) Magnified images of the inset in the top row. (**d**) Statistical analysis measuring γ-tubulin intensity around sDAPs (insets in panel [c]) in the WT and *CEP128*$^{-/-}$ centrioles (both WT and *CEP128*$^{-/-}$, n = 5 centrioles), **, p<0.01. (**e**) Two-color dSTORM images in panel (c) aligned and combined according to the longitudinal position of γ-tubulin relative to SCLT1 (n = 5 centrioles). (**f**) Statistical analysis of γ-tubulin intensity in the insets in panel (e) revealing that γ-tubulins are distributed towards the centriole distal end upon CEP128 depletion. (**g**) A model speculating on the role of sDAPs in microtubule anchoring. The loss of CEP128 relaxes the distribution of γ-tubulins toward the centriole distal end, whereas microtubules fail attach to the centriole at sDAPs upon CEP128 depletion. Bars = 200 nm.

of γ-tubulin at the sDAP tip, we compare the signals of γ-tubulin in the region potentially close to the sDAP tip between wild-type and $CEP128^{-/-}$ cells (*Figure 6c*). Averaging the signals of several centrioles shows statistically significant signal reduction in the sDAP tip region of $CEP128^{-/-}$ cells (*Figure 6d*). What is surprising is the difference of γ-tubulin occupancy along the mother centriole between wild-type and $CEP128^{-/-}$ cells (*Figure 6e and f*). Instead of being longitudinally confined to the level of the sDAPs shown in wild-type centrioles, the distribution of γ-tubulin in $CEP128^{-/-}$ centrioles spreads toward the distal end of the mother centriole close to the level of the DAPs. That is, sDAPs spatially regulate the distal boundary of γ-tubulins along the centriole. Furthermore, this population of γ-tubulins may not anchor MTs, because we observed decreased α-tubulin population at the distal end of the mother centriole in the $CEP128^{-/-}$ cells. as summarized in a speculative model in *Figure 5g*. Again, this result shows the presence of at least two populations of γ-tubulins in the $G_0$ phase. It is possible that a population of γ-tubulins at the sDAP tips is responsible for the nucleation of specific MTs from the sDAPs, although the specific functions of these MTs remain elusive.

## Discussion

We have mapped the molecular architecture of sDAPs together with DAPs using super-resolution microscopy to gain a more comprehensive structural understanding of the distal end of the mother centriole. Unlike the more consistent nine-fold symmetric structures of DAP proteins, sDAP proteins are less organized, either with an incomplete occupancy of the ring or a spreading radial distribution. For those sDAPs possessing a more complete ring-shaped occupancy, ODF2, CEP128, and centriolin form better-defined nine puncta-like structures, whereas CCDC68, ninein, and CEP170 distribute more randomly around a ring. ODF2 localizes close to the upper and lower ends of the root of an sDAP attaching to the centriolar MTs. The distal layer of ODF2 is downstream of the DAP protein CEP83, whereas the proximal layer of ODF2 is downstream of the sDAP protein CEP128. CEP128 localizes slightly outside of ODF2 radially and in between the two layers of ODF2 longitudinally. The genetic and localization relationships of ODF2 and CEP128 are consistent with the finding that these two proteins form a protein complex to work together as an sDAP component (*Kashihara et al., 2019*). Centriolin localizes slightly outside and distal to CEP128 in the central region of an sDAP. CEP89, in addition to its DAP layer, also has a longitudinal sDAP layer downstream of CEP128 localizing distal to centriolin, close to the distal border of an sDAP. Ninein and CEP170 both localize toward the tip of an sDAP and have relatively broad radial and longitudinal distributions. They potentially serve as key components for functions of sDAP tips, such as anchoring a subset of MTs around the mother centriole. Note that the measurement may differ depending on the epitope regions of the protein of interest, as observed using ODF2 antibodies that recognize each terminus of the protein (*Figure 4d*). This difference in structural arrangement may be related to the different functions of the N and C termini of ODF2 suggested in previous studies (*Ishikawa et al., 2005*; *Kashihara et al., 2019*), with the N terminus of ODF2 found to be the interacting domain for CEP128.

In addition to the dual roles of ODF2 and CEP89 on DAPs and sDAPs, we found that the longitudinal distribution of sDAP ninein is regulated by DAPs. Depleting DAPs results in the spreading of the ninein covering the longitudinal region from the sDAPs all the way toward the distal end of the mother centriole. That is, DAPs serve as a distal border of sDAPs. The structural arrangement of sDAPs is dependent on the structural arrangement of DAPs. Previously, the DAPs and the sDAPs were considered to be independent components serving distinct roles at the distal end of the mother centriole. Removal of sDAPs does not influence ciliogenesis, which requires proper functions of DAPs (*Mazo et al., 2016*). sDAP protein localizations are still observed upon depletion of DAPs (*Joo et al., 2013*). On the other hand, using super-resolution microscopy, we show here that DAPs and sDAPs are not entirely independent, with at least two proteins serving dual roles and with ninein positioning affected by the DAP structure (*Figure 4i*).

We also provide direct evidence that sDAP-specific CEP128 knockout depletes MT population close to the sDAPs, further confirming that sDAPs anchor a subset of MTs around the mother centriole (*Figure 6g*). This result clarifies the previous conclusion of ninein's role in MT anchoring as an sDAP element (*Delgehyr et al., 2005*; *Mogensen et al., 2000*). The exact functions of this subset of sDAP-anchoring MTs remains unclear due to the challenge of tracing individual MT filaments, even with our 20-nm super-resolution capability. One speculation would be related to

the mechanosensation of cells through this subset of sDAP-anchoring MTs. It is known that for a motile cilium, the component proximal to DAPs is a cone-shaped basal foot, whose orientation is aligned to its beating direction, serving as a mechanical coupling element at the ciliary base (*Clare et al., 2014*; *Bornens, 2012*; *Gibbons, 1961*). Because a primary cilium does not actively beat, it is speculated that mechanosensation is required in all orientations. Thus the ring-shaped sDAPs are formed at the centriolar distal end to anchor mechanosensing MTs in all circumferential directions. It would be interesting to examine mechanical coupling phenotypes in other types of polarized cells that have primary cilia, such as inner medulla collecting duct (IMCD) cells or Madin-Darby canine kidney (MDCK) cells. Another possibility is that this subset of MTs partially bridges a centriole and Golgi apparatus, because we have previously shown that even though Golgi remains adjacent to a centriole in single knockout of *CEP128*⁻/⁻, double knockouts of *CEP128*⁻/⁻ and *CNAP1*⁻/⁻ result in centriole-Golgi dissociation (*Mazo et al., 2016*). Because of the complex MT network surrounding the mother centriole, we were not able to determine whether the sDAP-linked MTs reach Golgi or not.

As γTuRC is the initiating template of a MT, we expected that the locations where γ-tubulins are populated would be where more α-tubulin signals are found if an MT nucleation site remains as an anchoring site. Surprisingly, their distributions around the mother centriole are inconsistent when observing them using super-resolution microscopy. In wild type cells, α-tubulin signals are mostly associated with DAPs and sDAPs, whereas γ-tubulins are mostly localized along the mother centriole below sDAPs. Although CEP128 depletion results in reduced α-tubulin close to the sDAP region, an unexpected increasing population of γ-tubulin is observed around the region between sDAPs and DAPs (*Figure 6g*). From these results, we can speculate that, in the G₀ quiescent phase, there is one population of γ-tubulins involved in MT anchoring at sDAPs, and another population of γ-tubulins localized along the centriole that is not associated with the α-tubulin population. The latter population comprises the majority of centrosomal γ-tubulins and wraps around the mother centriole at a well-defined cylindrical confinement with sDAPs as their distal constraint. It is possible that these γ-tubulins are anchored at the compact PCM structure outside the mother centriole, with a diameter larger than the centriole wall, illustrating a clear shape of the confined PCM during interphase. It is also possible that this major γ-tubulin population in G₀ phase serves as a reservoir of MT nucleating sites, which enables quick initiation of MT growth once the cells re-enter the cell cycle. Besides, as suggested by a previous study showing that MT initiation and anchoring may be independent processes (*Delgehyr et al., 2005*), it is also likely that these γ-tubulins can first initiate MT nucleation around the centriole and later be associated with some MT-anchoring proteins, such as ninein, to anchor MT to sDAPs. The function of this major γ-tubulin population may be interesting to address in future studies.

## Materials and methods

**Key resources table**

| Reagent type (species) or resource | Designation | Source or reference | Identifiers | Additional information |
|---|---|---|---|---|
| Cell line (*Homo-sapiens*) | hTERT RPE-1 | ATCC | CRL-4000 | Identity authenticated with STR Profiling by ATCC |
| Transfected construct (*Homo-sapiens*) | CEP83-Myc | *Lo et al., 2019* | | |
| Antibody | FBF1 (rabbit polyclonal) | Proteintech, Rosemont, IL, USA | 11531–1-AP | 1/200 |
| Antibody | SCLT1 (rat polyclonal) | *Tanos et al., 2013* | – | 1/250 |
| Antibody | CEP89 (rat polycloncal) | *Tanos et al., 2013* | – | 1/500 |

*Continued on next page*

*Continued*

| Reagent type (species) or resource | Designation | Source or reference | Identifiers | Additional information |
|---|---|---|---|---|
| Antibody | ODF2-N (rabbit polyclonal) | Sigma-Aldrich | HPA001874 | 1/200 |
| Antibody | ODF2-C (rabbit polyclonal) | Abcam | ab43840 | 1/200 |
| Antibody | CEP128 (rabbit polyclonal) | Abcam | ab118797 | 1/200 |
| Antibody | CENTRIOLIN (mouse monoclonal) | Santa Cruz | sc-365521 | 1/200 |
| Antibody | NINEIN (rabbit polyclonal) | Bethyl | A301-504 | 1/1000 |
| Antibody | NINEIN (mouse monoclonal) | Santa Cruz | sc-376420 | 1/500 |
| Antibody | CEP170 (rabbit polyclonal) | Abcam | ab72505 | 1/400 |
| Antibody | CCDC68 | Proteintech | 26301–1-AP | 1/400 |
| Antibody | C-NAP1 | Santa Cruz | sc-390540 | 1/200 |
| Antibody | γ-tubulin | Sigma-Aldrich | T6557 | 1/500 |
| Antibody | α-tubulin | Santa Cruz | sc-32293 | 1/500 |
| Antibody | Centrin (mouse monoclonal) | Millipore | 04–1624 | 1/400 |
| Recombinant DNA reagent | gRNA cloning vector | Addgene | #41824 | |
| Recombinant DNA reagent | CEP128 gRNA | (*Mazo et al., 2016*) | | gRNA2 (5'-GCTGCCAGATCAACGCACAGGG-3'), gRNA4 (5'-GAGTCAGCTCTGAGATCTGAAGG-3'), gRNA5 (5' GCAGCTGAACTTCAGCGCAATGG-3') |
| Recombinant DNA reagent | CEP83 gRNA | (*Mazo et al., 2016*) | | gRNA1 (5'-GGTGGAGACAGTGGATTGACAGG-3'), gRNA2 (5'-GATATTAACTCCACAAAAATTGG-3') |
| Software, algorithm | Metamorph | Molecular Device | | |
| Software, algorithm | ImageJ | NIH | | |

## Antibodies

The primary antibodies used in this study are listed in *Supplementary file 1* Table 2. Secondary antibodies used in this work were Alexa Fluor 488 (mouse A21202 and rabbit A21206; Thermo Fisher Scientific, Waltham, MA, USA), Alexa Fluor 647 (anti-mouse A21236, anti-rabbit A21245, anti-rat A21247; Thermo Fisher Scientific) and Cy3B-conjugated antibody, which was custom-made as described previously (*Yang et al., 2018*). Briefly, 10 mg/ml Cy3B maleimide (PA63131, GE Healthcare, Pittsburgh, PA, USA) dissolved in DMOS/DMF (1:1) was mixed with IgG antibodies (rabbit 711-005-152, rat 712-005-153; Jackson ImmunoResearch, West Grove, PA, USA) at a 1:1 ratio by volume. 0.67 M borate buffer (1859833, Thermo Fisher Scientific) was then added to the mixture, achieving a final concentration of 4%. The reaction mixture was protected from light and incubated at room temperature for 1 hr. The mixture was cleaned up using purification resin (1860513, Thermo Fisher Scientific) to remove excess dye and stored at 4°C until later use.

## Cell culture and immunofluorescence staining

Human retinal pigment epithelial cells (RPE-1) were purchased from ATCC (CRL-4000, Manassas, VA, USA) and were cultured in DMEM/F-12 medium (1:1; 11330–032, Gibco, Thermo Fisher Scientific, Waltham, MA, USA) supplemented with 10% fetal bovine serum (SH30109, Hyclone, GE Healthcare, Chicago, IL, USA) at 37°C with 5% $CO_2$. The identity of the cell line has been authenticated using STR Profiling Analysis by ATCC with a 100% match. A mycoplasma contamination test was performed by collecting cell culture medium without antibiotics for 24 hr and processing this medium by PCR using EZ-PCR mycoplasma detection kit (EZ-PCR Mycoplasma Detection Kit, 20-700-20, Biological Industries, CT, USA), according to the manufacturer's protocol. This cell line is not in the list of commonly misidentified cell lines maintained by the International Cell Line Authentication Committee. Prior to immunofluorescence staining, cells were cultured on poly-L-Lysine coated coverslips and then serum starved for 24 hr before fixing with methanol at −20°C. For staining of CCDC68, CEP170 and ninein, cells were first extracted with PTEM buffer for 2 min before proceeding to ice-cold methanol fixation. The PTEM buffer contained 20 mM PIPES, 0.2% Triton X-100, 10 mM EGTA and 1 mM $MgCl_2$ and was prepared at pH 6.8. After fixation, cells were permeabilized with 0.1% PBST (PBS with 0.1% Triton X-100) for 10 min before blocking with 3% BSA for 30 min. Primary antibodies at optimized dilution prepared with 0.1% BSA in PBST were then added to cells for 1 hr. To remove unbound antibodies, cells were washed three times with 0.1% PBST, and then incubated with optimally diluted secondary antibodies for 1 hr. Finally, cells were washed three times with 0.1% PBST and stored in PBS with sodium azide at 4°C until later use.

## Immunoblotting

Cells were washed with ice-cold PBS twice and lysed with 1% NP-40 buffer (20 mM Tris-HCl [pH 7.5], 150 mM NaCl, 1 mM EDTA, 1% NP-40 and 1x protease inhibitor cocktail [ROC-04693132001, Sigma-Aldrich]). Total cell lysates were incubated on ice for 20 min and centrifuged (14,000 x g, 10 min, 4°C). Following the protein quantification assay (500–0116, Bio-Rad, Hercules, CA USA), 30 μg proteins from wild-type and knockout cells were analyzed by 8% SDS-PAGE. Proteins were transferred to PVDF membrane and incubated with the indicated antibodies, before signals were detected using the Western Lightning Plus reagent (PK-NEL105, PerkinElmer, Waltham, MA, USA).

## CRISPR/Cas9-mediated generation of CEP83$^{-/-}$ and CEP128$^{-/-}$ cells

A CRISPR/Cas9 gene targeting technique was used to inactivate CEP83 or CEP128 in RPE-1 cells as described previously (*Mazo et al., 2016*). The targeting sequence of the gRNA was as followed: CEP128 gRNA2 (5′-GCTGCCAGATCAACGCACAGGG-3′), CEP128 gRNA4 (5′-GAGTCAGCTCTGAGATCTGAAGG-3′), CEP128 gRNA5 (5′ GCAGCTGAACTTCAGCGCAATGG-3′), CEP83 gRNA1 (5′-GGTGGAGACAGTGGATTGACAGG-3′), and CEP83 gRNA2 (5′-GATATTAACTCCACAAAAATTGG-3′). Multiple gRNAs targeting different exons for CEP128 were used to achieve complete protein depletion. Briefly, the targeting sequences were cloned into the gRNA cloning vector (#41824, Addgene, Cambridge, MA, USA) via the Gibson assembly method (New England Biolabs, Ipswich, MA, USA). gRNA was co-expressed with Cas9 protein in RPE-1 cells using reagents from the Church group (*Esvelt et al., 2013*) (Addgene: http://www.addgene.org/crispr/church/). Knockout cell lines were obtained through clonal propagation and then confirmed by genotyping and immunoblotting.

## Generation of CEP83$^{-/-}$Rescue cells

CEP83 construct synthesis and generation of CEP83$^{-/-}$ *Rescue* cells was as previously described *Lo et al. (2019)*. Briefly, *CEP83* cDNA amplified from a HELA cDNA library was first cloned into a pRK5M vector that tagged protein with Myc-epitope at the protein C-terminus. The CEP83-Myc sequence was subcloned into a pBabe-puro3 vector and then transfected into 293FT (R70007, Thermo Fisher Scientific) cells together with V-SVG and pCMV-gag-pol plasmids for lentivirus generation. The virus was collected 48 hr post infection. 3 ml of the viral stock was used to infect $5 \times 10^5$ CEP83$^{-/-}$RPE-1 cells seeded onto a 60 mm plate. Cells were selected 2 days after infection and maintained in culture medium containing 2 μg/ml puromycin (p8833, Sigma-Aldrich).

## Transmission electron microscopy

RPE-1 cells were grown on Aclar film (Electron Microscopy Sciences, Hatfield, PA, USA)-based coverslips and fixed in 4% paraformaldehyde (15710, Electron Microscopy Sciences) and 2.5% glutaraldehyde (G5882, Sigma-Aldrich, St. Louis, MO, USA) with 0.1% tannic acid in PBS buffer at 37°C for 30 min. Cells were then further postfixed in 1% $OsO_4$ in PBS buffer for 30 min on ice. After dehydrating in a graded series of ethanol, the cells were then infiltrated and embedded in EPON812 resin (Catalog #14120, Electron Microcopy Sciences) to generate a resin sample. A microtome (Ultracut UC6; Leica, Wetzlar, Germany) was used to cut the sample into serial sections (~90 nm thickness), which were then stained with 1% uranyl acetate and 1% lead citrate. Samples were imaged using FEI Tecnai Spirit G2 and a JOEL JEM-1400plus transmission electron microscope.

## dSTORM imaging and image analysis

For dSTORM imaging, Alexa Fluor 647 antibody (1:200) and Cy3B-conjugated antibody (1:100) were used as secondary antibodies. In general, the protein of interest was stained with Alexa Fluor 647 antibody whereas the reference protein, for example SCLT1, was stained with Cy3B-conjugated antibody. The dSTORM imaging system included a modified inverted microscope (Eclipse Ti-E, Nikon, Tokyo, Japan) and a laser merge module (ILE, Spectral Applied Research, Richmond Hill, Ontario, Canada) with individual controllers for three light sources. To illuminate samples in the wide field, photon beams from a 637 nm laser (OBIS 637 LX 140 mW, Coherent, Santa Clara, CA, USA), a 561 nm laser (Jive 561 150 mW, Cobolt, Solna, Sweden) and a 405 nm laser (OBIS 405 LX 100 mW, Coherent) were homogenized (Borealis Conditioning Unit, Spectral Applied Research) and focused with a 100 × 1.49 NA oil immersion objective (CFI Apo TIRF, Nikon). To minimize z-axis drift, a perfect focusing system (PFS, Nikon) was used. During dSTORM imaging, the 647 nm and 561 nm laser lines were operated at a high intensity of ~1–5 kW/cm$^2$ to quench most of the fluorescence from Alexa 647 and Cy3B, respectively. A weak 405 nm beam was used to activate a portion of the dyes, converting them from a dark state to an excitable state. For single-color imaging of Alexa 647, signals were filtered with a bandpass filter (700/75, Chroma, Bellows Falls, VT, USA); for two-color imaging, the Alexa 647 channel was first recorded, and then the Cy3B channel was acquired with the corresponding filter (593/40, Chroma).

The collected fluorescent signal was filtered through a single-band emission filter and acquired on an EMCCD (Evolve 512 Delta, Photometrics, Tucson, AZ, USA) with a pixel size of 93 nm. For each dSTORM image, 10,000–20,000 frames were acquired every 20 ms (exposure time). Individual single-molecule peaks were localized using MetaMorph Superresolution Module (Molecular Devices, Sunnyvale, CA, USA) based on a wavelet segmentation algorithm. Super-resolution images were cleaned with a Gaussian filter with a radius of 1 pixel. For imaging sample preparation, cells grown on coverslips were placed in an imaging chamber (Chamlide magnetic chamber, Live Cell Instrument, Seoul, Korea) and immersed in dSTORM imaging buffer. The buffer included TN buffer at pH 8.0, and an oxygen-scavenging system consisting of 60–100 mM mercaptoethylamine (MEA, 30070, Sigma-Aldrich) at pH 8.0, 0.5 mg/mL glucose oxidase (G2133, Sigma-Aldrich), 40 mg/mL catalase (C40, Sigma-Aldrich), and 10% (w/v) glucose (G8270, Sigma-Aldrich).

To correct lateral position drift, fiducial markers (Tetraspeck, T7279, Thermo Fisher Scientific) were added to the sample at a dilution of 1/200 before imaging. The drift was measured and corrected with ImageJ via frame-by-frame correlation of the fiducial markers. Chromatic aberration between the long and short wavelength channels was compensated with a customized algorithm relocating each pixel of a 561 nm image to its targeted position in the 647 nm channel with a predefined correction function obtained by a parabolic mapping of multiple calibration beads. For axial imaging of the DAPs, a ring pattern of SCLT1 was used to identify the top view orientation. For lateral imaging, the rod-like pattern of SCLT1 was used as an indicator of the lateral view of the centriole-cilium. To determine the diameter of the sDAP proteins, a radial position of an individual super-resolved punctum was measured and the distance between each punctum and the center was defined as the radius. To correlate dSTORM with EM images, signal of CEP170 was used to align with the tip of the sDAP contour on the electron micrograph.

## 3D model illustration

The 3D mother centriole structure model was drawn with the 3D illustration software Blender (Blender Foundation, Amsterdam, The Netherlands). The dimensions of the model were based on the mean localization positions measured from the dSTORM images (*Figure 1* and *Figure 2*) in this study, images from our previous work (*Yang et al., 2018*) and information from previous studies (*Anderson, 1972*).

## Expansion microscopy

For expansion microscopy, cells were fixed with 4% PFA (15710, Electron Microscopy Sciences) for 10 min before blocking. ATTO647N (40839, anti-rabbit-IgG, Sigma-Aldrich) and Alexa Fluor 488 secondary antibodies were used in immunostaining. Stained cells were then incubated in 0.1 mg/ml acryloyl-X (A20770, Thermo Fisher Scientific) solution in DMSO overnight at 4°C, followed by washing twice for 15 min each with PBS. For in-situ polymer synthesis, monomer solution (1x PBS, 2M NaCl, 8.6% sodium acrylate [408220, Sigma-Aldrich], 2.5% acrylamide [01697, Sigma-Aldrich], 0.15% N,N'-methylenebisacrylamide [M7279, Sigma-Aldrich]) was prepared as described previously (*Asano et al., 2018*). Samples were incubated with the monomer solution plus 0.2% ammonium persulfate (A3678, Sigma-Aldrich) and 0.2% TEMED (T7024, Sigma-Aldrich) at 4°C for 10 min in a wet chamber before transferred to 37°C for 1 hr for polymerization. Proteinase K (P8107S, New England Biolabs) was diluted 8 units/ml in digestion buffer containing 50 mM Tris-HCl (pH 8), 1 mM EDTA, 0.5% Triton X-100, 0.8 M NaCl and applied directly to gels for 1 hr at room temperature in the dark. After digestion, coverslips were removed carefully. For sample expansion, gels were immersed in water for 20 min. This step was repeated 2–3 times in fresh water until gels were totally expanded. The expanded gels were imaged with 1.49NA 100x objective operated in a spinning disk confocal mode. 3D stack was acquired every 200 nm in z axis. 3D images were rendered with ImageJ.

## Acknowledgements

This work was supported by the Ministry of Science and Technology (MOST), Taiwan 107–2112 M-001-037, 107–2313-B-001–009, and the Academia Sinica Career Development Award 2317–1040300 to WMC, T-JC, T-YC, Y-PL, TTY, and J-CL; by a MOST (108–2313-B-010–001, 108–2628-B-010–007, and 108–2638-B-010–001-MY2) award to W-JW and C-H L; and by an NIH grant GM088253 to M-FBT. This work was also supported in part by the Electron Microscopy Facility in NYMU.

## Additional information

### Funding

| Funder | Grant reference number | Author |
| --- | --- | --- |
| Ministry of Science and Technology, Taiwan | 107-2112-M-001-037 | Weng Man Chong<br>Tzu-Yuan Chiu<br>Ting-Jui Chang<br>You-Pi Liu<br>T. Tony Yang<br>Jung-Chi Liao |
| Ministry of Science and Technology, Taiwan | 107-2313-B-001-009 | Weng Man Chong<br>Tzu-Yuan Chiu<br>Ting-Jui Chang<br>You-Pi Liu<br>T. Tony Yang<br>Jung-Chi Liao |
| Academia Sinica | 2317-1040300 | Weng Man Chong<br>Tzu-Yuan Chiu<br>Ting-Jui Chang<br>You-Pi Liu<br>T. Tony Yang<br>Jung-Chi Liao |

| Ministry of Science and Technology, Taiwan | 108-2313-B-010-001 | Won-Jing Wang Chien-Hui Lo |
| Ministry of Science and Technology, Taiwan | 108-2628-B-010-007 | Won-Jing Wang Chien-Hui Lo |
| Ministry of Science and Technology, Taiwan | 108-2638-B-010-001 -MY2 | Won-Jing Wang |
| National Institutes of Health | GM088253 | Meng-Fu Bryan Tsou |

The funders had no role in study design, data collection and interpretation, or the decision to submit the work for publication.

### Author contributions

Weng Man Chong, Conceptualization, Resources, Data curation, Formal analysis, Validation, Investigation, Visualization, Methodology; Won-Jing Wang, Resources, Data curation; Chien-Hui Lo, Validation, Investigation, Visualization; Tzu-Yuan Chiu, Resources, Data curation, Methodology; Ting-Jui Chang, Data curation, Validation, Methodology; You-Pi Liu, Data curation, Methodology; Barbara Tanos, Gregory Mazo, Resources; Meng-Fu Bryan Tsou, Resources, Methodology; Wann-Neng Jane, Resources, Investigation; T Tony Yang, Conceptualization, Resources, Data curation, Formal analysis, Supervision, Investigation, Visualization, Methodology; Jung-Chi Liao, Conceptualization, Supervision, Funding acquisition, Investigation, Project administration

### Author ORCIDs

Weng Man Chong (iD) https://orcid.org/0000-0002-5548-1628
Won-Jing Wang (iD) http://orcid.org/0000-0001-9733-0839
Meng-Fu Bryan Tsou (iD) http://orcid.org/0000-0002-2159-8836
Jung-Chi Liao (iD) https://orcid.org/0000-0002-4323-6318

### Decision letter and Author response

Decision letter https://doi.org/10.7554/eLife.53580.sa1
Author response https://doi.org/10.7554/eLife.53580.sa2

## Additional files

### Supplementary files

• Supplementary file 1. Supplementary table 1. Radial and longitudinal positions of DAP and sDAP proteins. The positions of DAP and sDAP proteins from dSTORM images (relative to the DAP protein SCLT1) in the present study and our previous work (*Yang et al., 2018*). Supplementary table 2. Primary antibodies used in immunofluorescent staining. Summarized information on the sources, immunogens, and conditions of the primary antibodies used in the present study.

• Transparent reporting form

### Data availability

All data generated or analysed during this study are included in the manuscript and supporting files.

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
