## [Decision Letter]

**Acceptance summary:**

Centrioles are cylindrical microtubule-based structures that form the core of the major microtubule-organizing center, the centrosome, and also serve as basal bodies of cilia and flagella. Centriole functions partly depend of protrusions emerging from centriole surface, which are termed distal and subdistal appendages. This paper nicely combines super-resolution microscopy and CRISPR-Cas9 knockouts in human cells to reveal the molecular organization of centriole appendages. These data confirm previous observations and provide novel insight into the architecture of centriole appendages and also contribute to the ongoing discussion about the functional role and interdependence of these structures.

**Decision letter after peer review:**

Thank you for submitting your article "Super-resolution microscopy reveals coupling between mammalian centriole subdistal appendages and distal appendages" for consideration by *eLife*. Your article has been reviewed by three peer reviewers, one of whom is a member of our Board of Reviewing Editors, and the evaluation has been overseen by Anna Akhmanova as the Senior Editor. The following individuals involved in review of your submission have agreed to reveal their identity: Laurence Pelletier (Reviewer #3).

The reviewers have discussed the reviews with one another and the Reviewing Editor has drafted this decision to help you prepare a revised submission.

Summary:

In this manuscript, Wong and colleagues present imaging of sub-distal appendage (SDAP) architecture using quantitative dSTORM. The authors show that different SDAP proteins display different radial distributions around the mother centrioles. The authors then use similar analyses to map the lateral position occupied by each of the SDAP analysed and generate an axial and lateral composite view of SDAP organization, which suggests that ODF2 and CEP89 distribution are localized to both DAPs and SDAP region consistent with a potential positional relationship between these structures. The authors then go on to show that ODF2, CEP89 and Ninein distribution is differentially regulated upon the perturbation of DAP and SDAP integrity using CRISPR KO cell lines. The authors then demonstrate a small but reproducible reduction in the number of microtubules anchored in the vicinity of SDAP in CEP128 KO cells. Finally, the authors present data suggesting that the majority of γ-tubulin adjacent to the mother centriole does not appear to correlate with anchored microtubules at SDAPs in cells that are in G0 phase. Overall, this is a very nice paper, with high-quality quantitative sub-diffraction imaging of centriolar appendages. Although some of the data is somewhat preliminary, the results presented here expose both a positional and functional relationship between DAPs and SDAPs and begin to reveal a more direct role for SDAPs in microtubule anchoring at centrosomes, which will stimulate further work in these areas.

Essential revisions:

1) The most important problem with this work is that conclusions derived from STORM-based analyses are not corroborated by ultrastructural analysis.

Specifically, Figure 3 shows a 3D model of SDAPs. The model shows nine subdistal appendages. Since SDAPs occupy more than one microtubule triplet, the model presented in Figure 3 might not reflect the actual organization of SDAPs in RPE-1 cells. In addition, it is not obvious how this model was deduced from presented STORM data, which itself did not reveal a nine-fold distribution of most SDAP proteins. The authors will either need prove their model by performing EM analysis, or clarify that this is a speculative model. As it is now, it may suggest the wrong concept of SDAP architecture in mammalian cells. Please see PMID: 30045886, a recent review on this subject.

Further, in Figure 4I, which summarizes the localization changes of SDAP proteins in Cep128-/- and Cep83-/- cells, some type of appendage is drawn at the distal centriole end. Do authors mean to say that in the absence of distal appendages, subdistal appendages form at the centriole's very distal ends? If so, it would be critical to perform an EM analysis of wild type and knockout centrioles and to show whether there are any structural changes associated with centriole distal ends. Otherwise, it needs to be clear that the model is speculative. The cartoon in Figure 6G is not corroborated by STORM or ultrastructural analysis either.

In summary, it should be possible to perform some classical EM to, at minimum, to investigate whether in knockout cells lacking DAPs sub distal-like structures form at the distal ends of centrioles. If the models and cartoons cannot be proven by EM, then it would be essential to state very clearly that these models and cartoons are speculative.

2) Figure 4A-C. The authors show that after Cep128 and Cep83 knockout ODF2 STORM signal changes, losing its proximal portion after Cep128 removal. The same seems to occur after Cep83 depletion but it is not clear from Figure 4D which portion of ODF2 is lost after Cep83 depletion. However, based on 4B and C, there is a ~50 nm gap between Sclt-1 and ODF2 protein signals in Cep128-/- cells. Also, the brightest portion of the ODF2 signal is shifted for additional ~50 nm toward centriole proximal end. This is not consistent with the authors' interpretation of the data. Since it has been previously shown that ODF2 interacts with Cep128 via its N terminus which extends further away from centrioles, one interpretation why ODF2 signal changes in Cep128-/- cells could be that without its natural binding partner, it changes its configuration from more to less extended, changing the shape of the STORM signal.

3) The quality of some STORM images raises concerns. For instance, even a well characterized distal appendage protein SCLT-1, which is known to reproducibly localizes to nine discrete foci here on longitudinally analyzed centrioles shows a highly variable and irreproducible pattern. This raises a question about the reproducibility and reliability of other STORM signals, specifically in Figure 4G-H and 6C where Ninein and Γ-tubulin signals are very variable from one centriole to another. CP110 signal in Figure 4G is also of a poor quality, showing nonspecific signals. It is not clear how reliably it detects centriole distal ends. RPE-1 cells used in this study are supposed to be serum starved and, based on published data, majority of cells should have a cilium and should not have CP110 at centrioles.

Related to this: The authors use 24hours of FBS to synchronize cells in G0 and allow for, or not, cilia formation. Do the authors know how robust the G0 arrest and what % ciliation they achieve? It is unclear from looking at the data if they are imaging ciliated cells or not. More robust cell cycle arrest protocols could have been used or an additional cilia marker incorporated in their analysis to ensure differences observed in the +/- FBS conditions represents cell cycle modulation or the presence or not of a primary cilium.

4) Introduction: The authors claim that no clear EM image of SDAPs has been reported for human RPE-1 cells. This statement seems inaccurate, considering that there is a plethora of electron micrographs showing both sets of appendages in this cell line (for instance: PMID: 30988386, PMID: 23253480, PMID: 26675238, PMID: 25686250, PMID: 26880200... and more). It thus appears that SDAPs of RPE-1 cells are documented across literature appear to be present in variable number, that they can adopt various morphology (based on electron densities) and that they are associated with microtubules at their ends.

[Editors' note: further revisions were suggested prior to acceptance, as described below.]

Thank you for resubmitting your work entitled "Super-resolution microscopy reveals coupling between mammalian centriole subdistal appendages and distal appendages" for further consideration by *eLife*. Your revised article has been evaluated by Anna Akhmanova (Senior Editor) and two reviewers.

The manuscript has been improved but there are some remaining issues raised by reviewer 1 that need to be addressed before acceptance, as outlined below. Addressing these comments should not require any new experiments, but please take the various concerns and suggestions seriously and modify the paper accordingly. Please provide a point-by-point rebuttal explaining the changes that you have made.

Reviewer #1:

In the revised manuscript, the biggest issue remains the interpretation of the data and the model.

The authors have performed TEM analysis to understand the organization of subdistal appendages in starved RPE-1 cells (now shown in Figure 1I). They concluded that the TEM analysis supports the nine-ness in the organization of sDAS. However, there are several issues with this interpretation and, consequently, with the model.

Subsection 'ODF2 is close to the centriole microtubule wall whereas ninein and CEP170 are close to the sDAP tips' states that "The TEM images reveal that sDAP backbone can be mostly observed in the same section with a nine-fold arrangement". It appears that this conclusion was derived from only one cross-sectioned centriole (which is not adequate). The number of centrioles analyzed in cross section is not indicated But I saw only one analyzed centriole with eight instead of nine distinguishable sDA densities (marked by the asterix, one density which is detected in two adjacent sections is marked twice). So, in my view, presented electron micrograph clashes with author's interpretation. It demonstrates the lack of heterogeneity in the appearance of SDA in EM and questions the nine-fold arrangement of sDAs,

The authors explained (subsection 'Axial and lateral images compose a 3D molecular map of sDAPs' and rebuttal) that they used two studies to model SDAs. These are: Anderson et al., (PMID: 5064817), a study performed in cells from Rhesus monkey oviduct, and Paintrand et al., (PMID: 1486002), a study conducted on isolated centrioles from lymphoblastoid cell line.

Anderson study explores basal bodies which sport only one basal foot positioned in the middle of the basal body longitudinal axes. Moreover, the authors say that the "structure at the longitudinal position proximal to the DAPs is the basal foot, a 'badminton-shaped' structure largely different from the sDAPs in the mother centriole". It is, therefore, confusing how can Anderson model was used to model SDAs in RPE-1.

Paintrand study, another work used to model SDAs, describes centrioles isolated from lymphoblastoid cells. Since during centriole isolation the heads of SDAs are frequently lost, their conical appearance may as well be perturbed So, again, it is not clear how SDAs from this study can be taken as a benchmark for modeling od sDAs in intact cells. It is not obvious whether authors realize that most centriole electrographs presented in Paintrand study were additionally digitally modified (rotationally averaged) with the purpose to highlight specific discussion points. Such averaging will inevitably result in the appearance of nine subdistal appendages. Centrioles from the same study that were not rotationally averaged clearly show a variable number of sDAs and their more conical shape of sDA's densities (for instance see Figure 10.)

Therefore, the following interpretation of sDA morphology in the introduction paragraph needs to be changed. After all, author's own EM data is not in agreement with their own description.

Introduction: "When sDAPs do form a complete ring of a nine-fold symmetry, its symmetric pattern is different from that of DAPs (Uzbekov and Alieva, 2018), with one pairs of parallel electron-dense "spokes" in each arm, where one of the spokes is associated with the A-tubule of a MT triplet of the mother centriole and the other is associated with the C-tubule of an adjacent MT triplet...". It is also unclear why was Uzbekov and Alieva cited here, the centriole in question is taken over from Paintrand study.

EM analysis of SDAs of longitudinally positioned centrioles is now included in the manuscript (Figure 4J). It reveals variability in the shape of sDAs. All six appendages in wild type RPE cells are different in their morphology (not all have the same triangular shape) and they don't point in the same direction. It would be useful to accordingly adjust the interpretation in subsection 'Ninein covers a broadened longitudinal region toward the centriole distal end upon DAP removal'.

EM analysis of sDAs in Cep83-/- cells has revealed centrioles with somewhat uneven length and with sDAs positioned at various distances from both centriole's ends. Fewer sDA EM densities are also present in Cep83-\- cells. How this asymmetry is not reflected in the radial distribution of STORM Ninein signal, which remained more-less symmetrical organized around mother centriole? It is surprising that, given this revelation, the analysis of additional sDAPs has not been added to the analysis.

Introduction: "Loncarek group further used correlative super-resolution microscopy and EM to show precise localization of DAP proteins relative to the electron dense blade structure (Bowler et al., 2019), improving the architectural mapping of the DAPs". Please note that Bowler at al. used tomography to elucidate the organization of DA's electron densities, which argued against the notion that DAs are organized as blades.

The text says: "To understand the architecture of protein complexes at their mature stage in terms of structural occupancy, we further analyzed super-resolution images of those with nearly full ring occupancies for each sDAP protein.

What does it mean their mature stage? Were centrioles somehow pre-selected for imaging based on whether they saw a ring or not by wide field microscopy? Structural occupancy is unclear. I think that I understand the meaning, but the sentence could be re-phrased.

Arbitrary terminology seems to be used to describe the morphology of sDAs ("SDAP backbone", "sDAP arms", "spokes", "badminton-shaped, "root"...). It is confusing. sDA's parts are usually described as a head and a stem. At least these terms need to be defined.

Further proofreading is necessary. Some sentences are illogical; it is hard to understand what they meant. For instance: The title of the Figure 6 says: "γ tubulins around mother centriole do not nucleate MT anchoring at the sDAPs in the G0 phase". What does it mean that anchoring is not nucleated? The abbreviation "sDAP" appears erroneously used on some places. "sDAP protein" and "DAP protein" is also used, although "P" already stands for "protein"...

---

## [Author Response]

Essential revisions:1) The most important problem with this work is that conclusions derived from STORM-based analyses are not corroborated by ultrastructural analysis.Specifically, Figure 3 shows a 3D model of SDAPs. The model shows nine subdistal appendages. Since SDAPs occupy more than one microtubule triplet, the model presented in Figure 3 might not reflect the actual organization of SDAPs in RPE-1 cells. In addition, it is not obvious how this model was deduced from presented STORM data, which itself did not reveal a nine-fold distribution of most SDAP proteins. The authors will either need prove their model by performing EM analysis, or clarify that this is a speculative model. As it is now, it may suggest the wrong concept of SDAP architecture in mammalian cells. Please see PMID: 30045886, a recent review on this subject.

Originally we used TEM images of centrioles from previous studies (PMID: 5064817 by Anderson and PMID: 1486002 by Bornens' team) to deduce the architecture of the centriole and sDAP backbone as well as our dSTORM data to deduce the localization of sDAP proteins. Here to confirm the organization of sDAP backbone in RPE-1 cells, we have followsfollows the reviewer's suggestion and performed serial TEM sectioning of RPE-1 centrioles in the axial view, shown in Figure 1I. The TEM data reveals that sDAP backbone can be mostly observed in the same section with a nine-fold arrangement. It is also interesting to note that the sDAP arms are mostly wider than those of DAPs, similar to the observation recorded in Bornens' paper. Therefore, in our model, we constructed a nine-fold distribution of sDAPs. We have added this figure to the main text to support our model (Figure 1I). From the EM image, it seems that the arms of sDAPs are more flexible in shape and distribution than those of DAPs, potentially reflecting why the nine-fold symmetry is less obvious for the localization of sDAP proteins. We have stated the structural characteristics of the sDAP EM image as discussed above in the main text.

Our 3D model was constructed by combining measurements from previous studies, our current TEM results, and our dSTORM data, aiming to provide a view of the potential sDAP structure. Nonetheless, we also cannot rule out that sDAPs in RPE-1 cells may also be a dynamic structure as reported in PMID: 30045886, and therefore this sDAP structure may possibly reflect the organization of sDAPs at a subset of centrioles. We have thus edited the manuscript from subsection 'Axial and lateral images compose a 3D molecular map of sDA' to clarify this.

Further, in Figure 4I, which summarizes the localization changes of SDAP proteins in Cep128-/- and Cep83-/- cells, some type of appendage is drawn at the distal centriole end. Do authors mean to say that in the absence of distal appendages, subdistal appendages form at the centriole's very distal ends? If so, it would be critical to perform an EM analysis of wild type and knockout centrioles and to show whether there are any structural changes associated with centriole distal ends. Otherwise, it needs to be clear that the model is speculative. The cartoon in Figure 6G is not corroborated by STORM or ultrastructural analysis either.In summary, it should be possible to perform some classical EM to, at minimum, to investigate whether in knockout cells lacking DAPs sub distal-like structures form at the distal ends of centrioles. If the models and cartoons cannot be proven by EM, then it would be essential to state very clearly that these models and cartoons are speculative.

Thanks for the reviewer's comment. We have followsfollows the suggestion and performed TEM to examine the differences of sDAP structures between wild type and CEP83^-/-^ RPE-1 cells, as shown in Figure 4J. In wild type cells, sDAPs (or sub distal-like structure as mentioned by the reviewer) mostly maintain their longitudinal position, with the triangular arms pointing toward the direction close to perpendicular to the centriole orientation (Figure 4J, left panel). Surprisingly, we found that the positions and shapes of the arms of sDAPs are largely varying in CEP83 depleted cells, showing with more examples to illustrate the variations (Figure 4J, right panel). That is, the sDAP structure is much less stable when DAPs are missing. Specifically, a few examples show that sDAP tips may point toward the distal end with a thicker arm, consistent with what we observed in ninein super-resolution images of CEP83^-/-^ cells. This is in line with the previous TEM study of CEP164 knockdown RPE-1 centrioles (PMID: 23253480), where EM images show that sDAPs are titled toward the distal end. In other cases, sDAPs can localize at the middle region of the centriole, occupy the same side of the centriole, miss one side of occupancy, or localize at different longitudinal positions in the same centriole. These large variations imply that the loss of DAPs via CEP83 depletion affects the structural stability of sDAPs, reassuring that DAPs and sDAPs are structurally related to each other.

In accordance with these TEM results, we have revised our cartoon model for CEP83^-/-^ centrioles to demonstrate possible structural variations (Figure 4K).

2) Figure 4A-C. The authors show that after Cep128 and Cep83 knockout ODF2 STORM signal changes, losing its proximal portion after Cep128 removal. The same seems to occur after Cep83 depletion but it is not clear from Figure 4D which portion of ODF2 is lost after Cep83 depletion. However, based on 4B and C, there is a ~50 nm gap between Sclt-1 and ODF2 protein signals in Cep128-/- cells. Also, the brightest portion of the ODF2 signal is shifted for additional ~50 nm toward centriole proximal end. This is not consistent with the authors' interpretation of the data. Since it has been previously shown that ODF2 interacts with Cep128 via its N terminus which extends further away from centrioles, one interpretation why ODF2 signal changes in Cep128-/- cells could be that without its natural binding partner, it changes its configuration from more to less extended, changing the shape of the STORM signal.

We thank the reviewer for providing us insight into the potential change of ODF2 configuration upon CEP128 depletion. To further examine the idea, we used two different antibodies targeting both the N terminus and C terminus of ODF2 to examine their distribution differences in wild type and CEP128 depleted cells. The ODF2 antibody (HPA001874, Σ-Aldrich) we used in the manuscript targets the N terminus of ODF2, i.e. aa #39-200, out of its 829 amino acids, referred to as ODF2-N. We used another commercial antibody (ab43840, Abcam) that targets the C terminus of ODF2 (aa #800 to the C terminus), referred to as ODF2-C. dSTORM imaging of ODF2-C reveals that in wild type cells, ODF2-C distribution is wider than ODF2-N distribution in the radial direction (Figure 4B, 4C). In addition, ODF2-N extends more toward the proximal end than ODF2-C in the longitudinal direction, consistent with the CEP128 interaction result in PMID30623524. The distal edge of ODF2-C is closer to SCLT1 than that of ODF2-N, more reaching the centriole distal end. Interestingly, when CEP128 is depleted, distributions of both ODF2-N and ODF2-C become thinner. The gap between ODF2-N/C and SCLT1 of CEP128^-/-^ cells is larger than that of the wild type cells, illustrating both narrowing and shifting of ODF2 occupancy upon sDAP CEP128 depletion. These observations imply that CEP128, as the binding partner of ODF2, regulates the organization of ODF2. We have added these results to Figure 4B-4D in the main text (subsection 'ODF2 and CEP89 localizations are differentially regulated by DAP and sDAP integrity'). It remains to be examined what exactly the thinner layer of ODF2 in CEP128^-/-^ cells belongs to. One speculation is that the sDAP depletion results in the removal of ODF2 in the sDAP layer. We have stated clearly that this is only a speculation in the main text.

3) The quality of some STORM images raises concerns. For instance, even a well-characterized distal appendage protein SCLT-1, which is known to reproducibly localizes to nine discrete foci here on longitudinally analyzed centrioles shows a highly variable and irreproducible pattern. This raises a question about the reproducibility and reliability of other STORM signals, specifically in Figure 4G-H and 6C where Ninein and Γ-tubulin signals are very variable from one centriole to another. CP110 signal in Figure 4G is also of a poor quality, showing nonspecific signals. It is not clear how reliably it detects centriole distal ends. RPE-1 cells used in this study are supposed to be serum starved and, based on published data, majority of cells should have a cilium and should not have CP110 at centrioles.

In dual color dSTORM imaging, we used Alexa Fluor 647 (AF647) and Cy3b dyes for immunostaining. Out of the two dyes, AF647 performs better than Cy3b in terms of its blinking property, which provides images with a more homogeneous intensity. Therefore, we always mark our protein of interest with AF647 and use Cy3b as the reference marker. The aim of Figure 6B was to measure the diameter of γ-tubulin. Therefore, in Figure 6B, γ-tubulin was marked by AF647 dye while SCLT1 was used as a reference for the axial orientation. To illustrate the difference, here we have also imaged the SCLT1/γ-tubulin pair in a reverse manner, i.e. marking SCLT1 with AF647 and γ-tubulin with Cy3b as shown in Author response image 1, Right panel). It can be seen that SCLT1 shows a nine-fold distribution and γ-tubulin occupies a radial distribution similar to that in our original image.

**Author response image 1. sa2fig1:** Representative dSTORM images revealing the radial distribution of γ-tubulins. SCLT1 serves as a marker for an axial centriole view. (Left Panel) Figure 6b in the manuscript in which SCLT1 is marked by Cy3b dye and γ-tubulin by AF647. (Right panel) Staining of the SCLT1/γ-tubulin pair in a reverse manner, i.e. marking SCLT1 with AF647 dye and γ-tubulin with Cy3b. Bar = 200nm.

With regard to the imaging of CP110, we understand that serum starvation triggers ciliogenesis and CP110 should be absent on ciliated centrioles. We chose to use 24-hour starvation on purpose because it will give us ~60% ciliated cells (as we previously reported in PMID31455668, Author response image 2), allowing us to find centrioles both perpendicular and parallel to the glass surface and thus to perform axial and lateral imaging, respectively. Since we lacked the location reference marker of DAPs in CEP83 depleted cells that we usually used, in order to compare the longitudinal distribution of ninein between wild type and CEP83 depleted cells, we thus searched for laterally oriented centrioles with CP110 to enable longitudinal localization of proteins of interest.

**Author response image 2. sa2fig2:** A bar graph comparing the percentage of ciliated RPE-1 cells after serum starvation (SF) for 24 hour (1 day) and 48 hours (2 days). Figure excerpted from figure 1C of our previous work (PMID31455668).

CP110 is present on daughter and mother centrioles, which would be seen as two dots in close proximity to each other. To distinguish mother centriole CP110 from that of the daughter centriole, we used both centrin and ninein as a marker. For a laterally oriented mother centriole, centrin is observed in wide field imaging as a rod-shaped structure (centriole) with ninein at the sDAP region perpendicular to the centriole, whereas CP110 on a mother centriole localizes at the edge of the centriole 'rod' close to ninein at sDAP region (as shown in Author response image 3). For our dSTORM imaging, we used this method to help us determine the distal end of a mother centriole.

**Author response image 3. sa2fig3:** Wide field imaging of centriole in the lateral view. Bar = 2um.

To reassure the validity of our results, here we have also used CEP97 as the distal end marker. As compared to wild type cells, ninein distribution in CEP83^-/-^ cells is more relaxed and extended (as shown in Author response image 4). Ninein seems to localize in a closer proximity to CEP97 in CEP83^-/-^ cells as compared to in wild type cells. This data is consistent with our dSTORM data using CP110 as the distal end marker.

**Author response image 4. sa2fig4:** Dual Color dSTORM imaging of Ninein and CEP97. Bars = 200nm.

Related to this: The authors use 24hours of FBS to synchronize cells in G0 and allow for, or not, cilia formation. Do the authors know how robust the G0 arrest and what % ciliation they achieve? It is unclear from looking at the data if they are imaging ciliated cells or not. More robust cell cycle arrest protocols could have been used or an additional cilia marker incorporated in their analysis to ensure differences observed in the +/- FBS conditions represents cell cycle modulation or the presence or not of a primary cilium.

We have performed flow cytometry for cycling, 24-hour and 48-hour serum starved RPE-1 cells to study the cell population for each cell phase. We found that both 24 hours and 48 hours of serum starvation drove most of the cells to the G0/G1 population (Author response image 5). In terms of ciliation, we studied the ciliation frequencies upon 24-hour and 48-hour serum starvation in our previous work (PMID31455668), which were around 60% and 80%, respectively (Author response image 2).

In our study, we need to image both laterally oriented and axially oriented centrioles. From our experience, we found that centrioles in RPE-1 cells starved for 48 hours were mostly laterally oriented, whereas cells starved for 24 hours showed centrioles in various orientations. We therefore chose 24-hour starvation as the time point for our study. In addition, we used this time point for all our images in this work to maintain consistency of the data.

**Author response image 5. sa2fig5:** Flow cytometry analysis of 0, 24, 48-hour serum starved RPE-1 cells. This data reveals that 24-hour and 48-hour serum starvations drive most cells to the G0/G1 population.

4) Introduction: The authors claim that no clear EM image of SDAPs has been reported for human RPE-1 cells. This statement seems inaccurate, considering that there is a plethora of electron micrographs showing both sets of appendages in this cell line (for instance: PMID: 30988386, PMID: 23253480, PMID: 26675238, PMID: 25686250, PMID: 26880200... and more). It thus appears that SDAPs of RPE-1 cells are documented across literature appear to be present in variable number, that they can adopt various morphology (based on electron densities) and that they are associated with microtubules at their ends.

We thank the reviewer for the advice. We have removed the sentence 'no clear EM image of SDAPs has been reported for human RPE-1 cells' in the Introduction to avoid confusion. The list of articles suggested by the reviewer includes TEM images of PRE-1 centriole in the lateral view upon CEP120 knockout (PMID: 30988386), CEP164 knockdown (PMID: 23253480), WDR8 knockdown (PMID: 26675238), EHD1 knockdown (PMID: 25686250) and NEDD1 overexpression (PMID: 26880200). CEP164 is a DAP protein among these proteins. Interestingly, from the siCEP164 EM images (PMID: 23253480), we also observed tilted sDAP structure similar to that of our CEP83 depleted RPE-1 centrioles (Figure 1G of PMID: 23253480). This data echoes with our observation that loss of DAP affects sDAP stability, suggesting a structural relationship between them.

[Editors' note: further revisions were suggested prior to acceptance, as described below.]

Reviewer #1:In the revised manuscript, the biggest issue remains the interpretation of the data and the model.The authors have performed TEM analysis to understand the organization of subdistal appendages in starved RPE-1 cells (now shown in Figure 1I). They concluded that the TEM analysis supports the nine-ness in the organization of sDAS. However, there are several issues with this interpretation and, consequently, with the model.Subsection 'ODF2 is close to the centriole microtubule wall whereas ninein and CEP170 are close to the sDAP tips' states that "The TEM images reveal that sDAP backbone can be mostly observed in the same section with a nine-fold arrangement". It appears that this conclusion was derived from only one cross-sectioned centriole (which is not adequate). The number of centrioles analyzed in cross section is not indicated but I saw only one analyzed centriole with eight instead of nine distinguishable sDA densities (marked by the asterix, one density which is detected in two adjacent sections is marked twice). So, in my view, presented electron micrograph clashes with author's interpretation. It demonstrates the lack of heterogeneity in the appearance of SDA in EM and questions the nine-fold arrangement of sDAs,

We thank the reviewer for the comments. The serial cross-sectioned TEM images in Figure 1I are from one centriole (as shown in Figure Author response image 6). To further elaborate our finding, we have reassured the alignment of images Z4 and Z5 using the original EM images as shown in Figure Author response image 6. By further aligning the nine triplets of the centriole, we find that the missing sDAP in image Z4 between two adjacent triplets is indeed present in image Z5. We have stacked the serial TEM images and removed interference from the DAP signal to form a composite image and illustrate the nine sDAP EM signals (Author response image 6). We have also performed an additional centriole TEM sectioning as shown in Figure Author response image 7. After stacking and removing signals from DAP, we also observed nine sDAPs in this centriole. We agree with the reviewer and a recent review about sDAPs (Uzbekov and Alieva, 2018) that sDAPs are dynamic and varying structures, especially in the number of appendages as we mentioned in the Introduction that the range of 2-12 sDAPs has been reported. We do not intend to use our TEM images to claim a consistent nine-fold organization of sDAPs as that of the DAPs, as we have shown that in the beginning of the Results section (Figure 1A-1F). Our 3D model represents one potential organization of sDAPs, assuming the condition when the sDAP region is fully occupied and each sDAP stem is found between two microtubule triplets. We have also emphasized as in subsection 'Axial and lateral images compose a 3D molecular map of sDAPs' that our model is an example for the potential organization of a subset of sDAPs with the sentence 'Note that because the structures of sDAPs are dynamic (Uzbekov and Alieva, 2018), this model only represents one possible organization of sDAPs in a subset of centrioles, such that other settings may also exist.'

**Author response image 6. sa2fig6:** Serial TEM imaging of a mother centriole in a wild type RPE-1 cell from Figure 1I. (A) The serial TEM sections from Figure 1I. Longitudinal difference between each plane is around 90 nm (B) TEM images of the Z4 and Z5 planes assuring alignment of the images. Similarity in background signals between the two planes are indicated by dashed line and arrows. (C) (Left top panel) A cartoon illustrating the orientation of the Z4 and Z5 planes of the centriole; a dashed line represents the z position of the TEM section. (Left middle panel) Rotational alignment of MT triplets in Z5 plane to that of the Z4 plane; the nine DAPs are indicated by numbers in green. (Left bottom panel) Removal of DAP signal (marked by red circular shapes) in the Z5' image. (Right) Stacking of Z4 and Z5'' images to reveal the position of the nine sDAP stems as indicated by numbers in red. Bars = 100 nm.

**Author response image 7. sa2fig7:** Serial TEM imaging of a mother centriole in a wild type RPE-1 cell. The cartoon in the upper panel illustrates the orientation of the centriole; a dashed line represents the z position of each TEM section (Z1-Z6, from proximal to centriole distal end) with a longitudinal difference of around 90 nm each. The Z4' image is derived from Z4 with the DAP signal removed. The composite image composed of Z2, Z3 and Z4' reveals the presence of nine sDAP stems. Bar = 100 nm.

The authors explained (subsection 'Axial and lateral images compose a 3D molecular map of sDAPs'and rebuttal) that they used two studies to model SDAs. These are: Anderson et al., (PMID: 5064817), a study performed in cells from Rhesus monkey oviduct, and Paintrand et al., (PMID: 1486002), a study conducted on isolated centrioles from lymphoblastoid cell line.Anderson study explores basal bodies which sport only one basal foot positioned in the middle of the basal body longitudinal axes. Moreover, the authors say that the "structure at the longitudinal position proximal to the DAPs is the basal foot, a 'badminton-shaped' structure largely different from the sDAPs in the mother centriole". It is, therefore, confusing how can Anderson model was used to model SDAs in RPE-1.Paintrand study, another work used to model SDAs, describes centrioles isolated from lymphoblastoid cells. Since during centriole isolation the heads of SDAs are frequently lost, their conical appearance may as well be perturbed So, again, it is not clear how SDAs from this study can be taken as a benchmark for modeling od sDAs in intact cells. It is not obvious whether authors realize that most centriole electrographs presented in Paintrand study were additionally digitally modified (rotationally averaged) with the purpose to highlight specific discussion points. Such averaging will inevitably result in the appearance of nine subdistal appendages. Centrioles from the same study that were not rotationally averaged clearly show a variable number of sDAs and their more conical shape of sDA's densities (for instance see Figure 10.)Therefore, the following interpretation of sDA morphology in the introduction paragraph needs to be changed. After all, author's own EM data is not in agreement with their own description.

We thank the reviewer for the comments. The two models from Anderson's TEM study and Paintrand's study, which we used as references for our 3D centriole model, were actually used to construct (1) the centriole wall with the distal appendages and (2) the subdistal appendages, respectively. The same as we mentioned above, we agree with the reviewer that the number of sDAP varies as shown in Figure 9 of Paintrand's paper Figure , in which a range of around 3-9 sDAPs can be observed. We also realize that some of the images in Paintrand's study were rotationally averaged. However, the presence of nine sDAP stems was also observed in a few of the non-averaged images as shown in Figure 3 Figure and Figure 9 from Paintrand's work. Again, we do not intend to emphasize a consistent nine-fold arrangement of sDAP. Our model represents one of the potential arrangements existing in a subset of sDAPs, as we clarified in subsection 'Axial and lateral images compose a 3D molecular map of sDAPs'. The main goal of constructing a 3D model is to provide a schematic view on the potential positioning of sDAP proteins on the mother centriole.

To avoid confusion, we have edited the sentences to further clarify the construction of our 3D model and emphasize the dynamic nature of sDAP structure starting from subsection 'Axial and lateral images compose a 3D molecular map of sDAPs' as follows: "The model of the centriole and the DAPs are based on previous TEM serial sections of the monkey oviduct basal body (Anderson, 1972); the sDAP model is constructed based on the TEM study of centrioles from a human lymphoblastoma cell line (Paintrand et al., 1992) and our sDAP TEM results (Figure 1I). Note that because the structures of sDAPs are dynamic (Uzbekov and Alieva, 2018), this model only represents one possible organization of sDAPs in a subset of centrioles, such that other settings may also exist."

Introduction: " When sDAPs do form a complete ring of a nine-fold symmetry, its symmetric pattern is different from that of DAPs (Uzbekov and Alieva, 2018), with one pairs of parallel electron-dense "spokes" in each arm, where one of the spokes is associated with the A-tubule of a MT triplet of the mother centriole and the other is associated with the C-tubule of an adjacent MT triplet...". It is also unclear why was Uzbekov and Alieva cited here, the centriole in question is taken over from Paintrand study.

With regard to the reference, the reason we cite the article (Uzbekov and Alieva, 2018) is because it is the first comprehensive review paper comparing and summarizing the difference between sDAPs and DAPs, supporting our statement that the symmetric pattern of sDAPs is different from that of DAPs. We have also included Paintrand work as a reference for nine-fold symmetric morphology of the sDAPs.

To clarify, we have edited the Introduction as follows:

"Some EM images showed variations in the number of sDAPs, such as 2 to 12 sDAP stems in human endotheliocytes (Bystrevskaya et al., 1992, Bystrevskaya et al., 1988), illustrating the dynamic nature of sDAPs. That is, in contrast to the exact number of nine DAPs per centriole, the number of sDAPs may be different in different centrioles (Uzbekov and Alieva, 2018). Even when sDAPs do form a complete ring of a nine-fold symmetry, its morphology is different from that of DAPs (Paintrand et al., 1992, Uzbekov and Alieva, 2018). Each sDAP stem is composed of one pair of electron-dense signals on the sides, where one of them is associated with the A-tubule of a MT triplet of the mother centriole and the other is associated with the C-tubule of an adjacent MT triplet (Bystrevskaya et al., 1988, Paintrand et al., 1992)."

EM analysis of SDAs of longitudinally positioned centrioles is now included in the manuscript (Figure 4J). It reveals variability in the shape of sDAs. All six appendages in wild type RPE cells are different in their morphology (not all have the same triangular shape) and they don't point in the same direction. It would be useful to accordingly adjust the interpretation in subsection 'Ninein covers a broadened longitudinal region toward the centriole distal end upon DAP removal'.

We thank the reviewer's suggestion and have changed the sentence in subsection 'Ninein covers a broadened longitudinal region toward the centriole distal end upon DAP removal' as follows:

"In wild type cells, sDAPs mostly maintained their longitudinal position close to and proximal to the DAPs, but their shapes varied in different centrioles or even in the same centriole."

EM analysis of sDAs in Cep83-/- cells has revealed centrioles with somewhat uneven length and with sDAs positioned at various distances from both centriole's ends. Fewer sDA EM densities are also present in Cep83-\- cells. How this asymmetry is not reflected in the radial distribution of STORM Ninein signal, which remained more-less symmetrical organized around mother centriole? It is surprising that, given this revelation, the analysis of additional sDAPs has not been added to the analysis.

Comparing our EM analysis of Cep83^-/-^ cells with our dSTORM images for longitudinal distribution of ninein (Figure 4H, middle panel), we agree with the reviewer that the ninein dSTORM signal does not show a high level of asymmetry. This is related to our centriole pre-selection process prior to dSTORM imaging. Since the DAP proteins are absent in CEP83^-/-^ cells, SCLT1 or any other large diameter DAP protein cannot be used as a marker for the centriole orientation. Therefore, when we imaged ninein in CEP83^-/-^ cells, ninein itself was also used as an orientation marker. In our experiment, we co-stained CP110, ninein and centrin in CEP83^-/-^ cells as shown in Figure Author response image 8. To identify a laterally oriented mother centriole, centrin is observed in widefield imaging as a rod-shaped structure (centriole) with ninein at the sDAP region perpendicular to the centriole; while CP110 on a mother centriole localizes at the edge of the centriole 'rod' close to ninein at sDAP region, serving as a reference for the distal end. With this pre-selection process, the ninein that lacks either left or right signal at the sDAP region (similar to those CEP83^-/-^ cells that lack one of the electron density signals at sDAPs) would not be selected because it was challenging to assure that these centrioles were parallel to the imaging plane and thus to image them from a lateral view point. The ninein selected for dSTORM imaging would be the one with both left and right signals. Indeed, in the dSTORM imaging, the ninein signal is less asymmetric (Figure 4H, middle panel), but their distribution pattern is comparable to one of the EM images in CEP83^-/-^ cells (Figure 4J, right panel) in which sDAPs are present on both sides and occupy a wider longitudinal space than those of the wild type cells. Thus, our ninein images reflect its distribution in a subset of centrioles, and this centriole pre-selection process should be the reason why the left-right asymmetry is less obvious in dSTORM images of ninein. To clarify this, we have added a sentence in subsection 'Ninein covers a broadened longitudinal region toward the centriole distal end upon DAP removal' as follows: "Note that laterally oriented centrioles were preselected by the pattern of centrin and ninein under widefield imaging prior to dSTORM imaging; therefore, unlike the asymmetry of sDAP stems observed in the EM images of CEP83^-/-^ cells (Figure 4J, right panel), asymmetry of ninein is less obvious in the ninein super-resolution images of CEP83^-/-^ cells (Figure 4H)."

**Author response image 8. sa2fig8:** Widefield imaging of a centriole in the lateral view. Bar = 2um.

Introduction: "Loncarek group further used correlative super-resolution microscopy and EM to show precise localization of DAP proteins relative to the electron dense blade structure (Bowler et al., 2019), improving the architectural mapping of the DAPs". Please note that Bowler at al. used tomography to elucidate the organization of DA's electron densities, which argued against the notion that DAs are organized as blades.

We thank the reviewer's comment. Bowler's paper suggested that each DAP is anchored to the centriole by a triangle shaped base with their centriole EM images, and indeed only stated that their finding of the triangle shaped base "gives a perception of a 'blade'". To avoid confusion, we changed the sentence in the Introduction as follows: "Loncarek group further used correlative super-resolution microscopy and EM to show precise localization of DAP proteins relative to the electron dense structure of DAPs (Bowler et al., 2019), improving the architectural mapping of the DAPs."

The text says: "To understand the architecture of protein complexes at their mature stage in terms of structural occupancy, we further analyzed super-resolution images of those with nearly full ring occupancies for each sDAP protein.What does it mean their mature stage? Were centrioles somehow pre-selected for imaging based on whether they saw a ring or not by wide field microscopy? Structural occupancy is unclear. I think that I understand the meaning, but the sentence could be re-phrased.

When we mapped the radial distribution of sDAP proteins, we purposely searched for the ring-like structure of sDAP proteins by widefield microscopy prior to dSTORM imaging. That is, in the reviewer's word, a subset of centrioles was pre-selected based on whether we saw a ring or not by widefield microscopy. Our goal was to understand the radial arrangement of sDAP proteins when they formed a ring-like structure. The term 'mature stage' was used to define such centrioles. To clarify, we have rephrased the sentences in subsection 'ODF2 is close to the centriole microtubule wall whereas ninein and CEP170 are close to the sDAP tips' as follows.

"To understand the architecture of sDAP proteins with nearly full occupancies at the sDAPs, we pre-selected centrioles based on whether we saw a ring or not for an sDAP protein of interest by widefield microscopy and further analyzed them with super-resolution microscopy."

Arbitrary terminology seems to be used to describe the morphology of sDAs ("SDAP backbone", "sDAP arms", "spokes", "badminton-shaped, "root"...). It is confusing. sDA's parts are usually described as a head and a stem. At least these terms need to be defined.

We used 'DAP arm' as in the Boren's paper (Paintrand et al., 1992) to represent a single sDAP stem, 'DAP backbone' to describe the entire sDAP structure on the mother centriole and 'spokes' to refer to the electron dense signal on each side of a sDAP stem. To avoid confusion, we have adapted the term 'sDAP stem(s)' for sDAP arm or backbone and removed the term 'spokes'.

The term 'root' means the portion of a sDAP stem which connects the centriole wall. We have further clarified its definition when it first appeared in subsection 'Lateral super-resolution images reveal sDAP as a triangular structure'.

Further proofreading is necessary. Some sentences are illogical; it is hard to understand what they meant. For instance: The title of the Figure 6 says: "γ tubulins around mother centriole do not nucleate MT anchoring at the sDAPs in the G0 phase". What does it mean that anchoring is not nucleated?

We thank the reviewer for the reminder. To avoid confusion, we have corrected the title of Figure 6 to "A majority of γ-tubulins around the mother centriole are not associated with MT anchoring at sDAPs in the G0 phase". We have also proofread the entire manuscript carefully and made several changes in the manuscript.

The abbreviation "sDAP" appears erroneously used on some places. "sDAP protein" and "DAP protein" is also used, although "P" already stands for "protein"...

We also noticed that various short forms exist in the literature for distal appendages (e.g. DA, DAP, DAPs) and subdistal appendages (e.g. sDA, SDA, sDAP, sDAPs). In our work, we used sDAP and DAP as the abbreviation for subdistal appendage and distal appendage, respectively, following papers such as PMID: 24882706, PMID: 24231678 and PMID: 29514088. These terms are defined when they first appear in the Introduction.